# MathFimer: Enhancing Mathematical Reasoning by Expanding Reasoning Steps through Fill-in-the-Middle Task

**Yuchen Yan[1,2]**\*, **Yongliang Shen[1]**†, **Yang Liu[2]**, **Jin Jiang[2,3]**, **Xin Xu[4]**,
**Mengdi Zhang[2]**, **Jian Shao[1]**†, **Yueting Zhuang[1]**
[1]Zhejiang University [2]Meituan Group [3]Peking University
[4]Hong Kong University of Science and Technology
{yanyuchen,syl,jshao}@zju.edu.cn

## Abstract

Mathematical reasoning represents a critical frontier in advancing large language models (LLMs). While step-by-step approaches have emerged as the dominant paradigm for mathematical problem-solving in LLMs, the quality of reasoning steps in training data fundamentally constrains the performance of the models. Recent studies have demonstrated that more detailed intermediate steps can enhance model performance, yet existing methods for step expansion either require more powerful external models or incur substantial computational costs. In this paper, we introduce MathFimer, a novel framework for mathematical reasoning step expansion inspired by the "Fill-in-the-middle" task from code reasoning. By decomposing solution chains into prefix-suffix pairs and training models to reconstruct missing intermediate steps, we develop a specialized model, MathFimer-7B, on our carefully curated NuminaMath-FIM dataset. We then apply these models to enhance existing mathematical reasoning datasets by inserting detailed intermediate steps into their solution chains, creating MathFimer-expanded versions. Through comprehensive experiments on multiple mathematical reasoning datasets, including MathInstruct, MetaMathQA and etc., we demonstrate that models trained on MathFimer-expanded data consistently outperform their counterparts trained on original data across various benchmarks such as GSM8K and MATH. Our approach offers a practical, scalable solution for enhancing mathematical reasoning capabilities in LLMs without relying on powerful external models or expensive inference procedures.

## 1 Introduction

Recent advances in large language models (LLMs) (OpenAI et al., 2024; DeepSeek-AI et al., 2025) have demonstrated remarkable capabilities across various reasoning tasks (Gao et al., 2024; Xu et al., 2025), from logical deduction to complex problem-solving (Phan et al., 2025). Among these, mathematical reasoning stands as a particularly challenging frontier (Sun et al., 2025; Xu et al., 2024), serving as a critical benchmark for evaluating an LLM's ability to perform structured, multi-step reasoning processes.

A key breakthrough in improving LLMs' mathematical reasoning capabilities has been the introduction of chain-of-thought (CoT) prompting (Wei et al., 2022), where models explicitly articulate intermediate steps in their problem-solving process. This approach has not only enhanced solution accuracy but has also provided valuable insights into the models' reasoning mechanisms. However, the effectiveness of CoT prompting raises a fundamental question: ***What characteristics of training data are crucial for developing LLMs that can generate high-quality reasoning chains and arrive at correct mathematical solutions?***

---

\*Contribution during internship at Meituan Group.
†Corresponding authors.

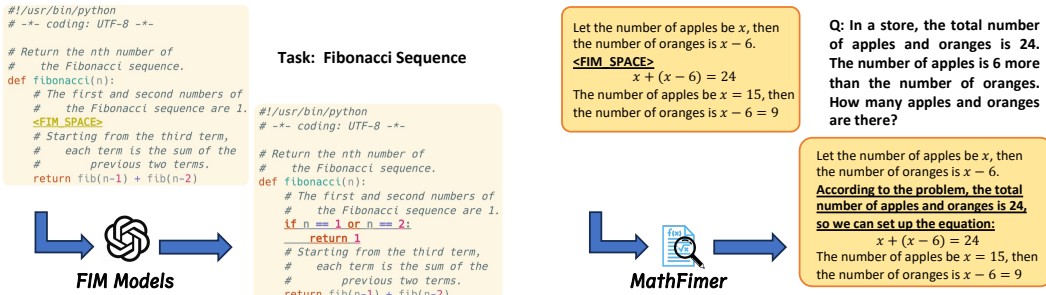

(a) FIM models in code reasoning.          (b) MathFimer in mathematical reasoning.

Figure 1: We developed MathFimer inspired by the fill-in-the-middle task in code reasoning of LLMs. Panel 1a demonstrates an example where the FIM model completes a given code context, while Panel 1b shows how MathFimer, as proposed in this paper, extends the steps of an existing step-by-step answer.

Prior research has revealed that the granularity and completeness of reasoning steps in training data significantly impact a model's reasoning capabilities (Jin et al., 2024). Models trained on more detailed step-by-step solutions tend to exhibit superior performance in mathematical reasoning tasks. This observation has led to various approaches for expanding reasoning steps in training data, including the use of stronger external models and sophisticated search algorithms like Monte Carlo Tree Search (MCTS) (Zhou et al., 2024a; Wu et al., 2024; Liu et al., 2024). However, these current approaches to improving reasoning steps face three main challenges. First, they rely on using even larger models to create better steps, which creates a cycle where we constantly need bigger models to make improvements (Guan et al., 2025; Toshniwal et al., 2024). Second, these methods require substantial computing resources, particularly when using advanced techniques like MCTS to explore different reasoning paths. Third, instead of building upon existing human-verified steps, these methods often generate entirely new reasoning chains, which can introduce unexpected errors and reduce the reliability of solutions.

These limitations motivate our central research question: ***Can we develop a more efficient and reliable method for expanding reasoning steps while preserving the validity of existing human-generated solutions?*** Drawing inspiration from the "fill-in-the-middle" task in code reasoning (Bavarian et al., 2022), where LLMs successfully complete missing code segments based on surrounding context, we propose a novel approach to this problem. Rather than generating entirely new reasoning chains, we explore whether the FIM paradigm can be adapted to supplement missing steps in existing reasoning chain or insert more detailed explanations into already sufficient steps.

Building on this insight, we propose MathFimer, a framework for enhancing mathematical reasoning through step expansion. We first construct NuminaMath-FIM by decomposing NuminaMath-CoT (Li et al., 2024) solutions into prefix-suffix pairs with missing intermediate steps. Using this dataset, we train a step-expansion model MathFimer-7B on math-specialized base model Qwen2.5-Math-7B (Yang et al., 2024). This model learns to supplement intermediate reasoning steps while preserving the original solution structure.

We apply MathFimer-7B to expand the reasoning steps in several existing mathematical reasoning datasets and evaluate their impact through comprehensive experiments. Our results demonstrate that training on MathFimer-expanded data consistently improves model performance across various mathematical reasoning benchmarks, including GSM8K and MATH. This improvement is observed across both general-purpose and math-specialized models, with expanded datasets leading to more detailed reasoning steps and higher solution accuracy compared to the original training data.

Our main contributions are threefold:

- We propose a novel step expansion framework inspired by code completion techniques, introducing MathFimer to enhance mathematical reasoning through targeted insertion of intermediate steps in existing solutions.

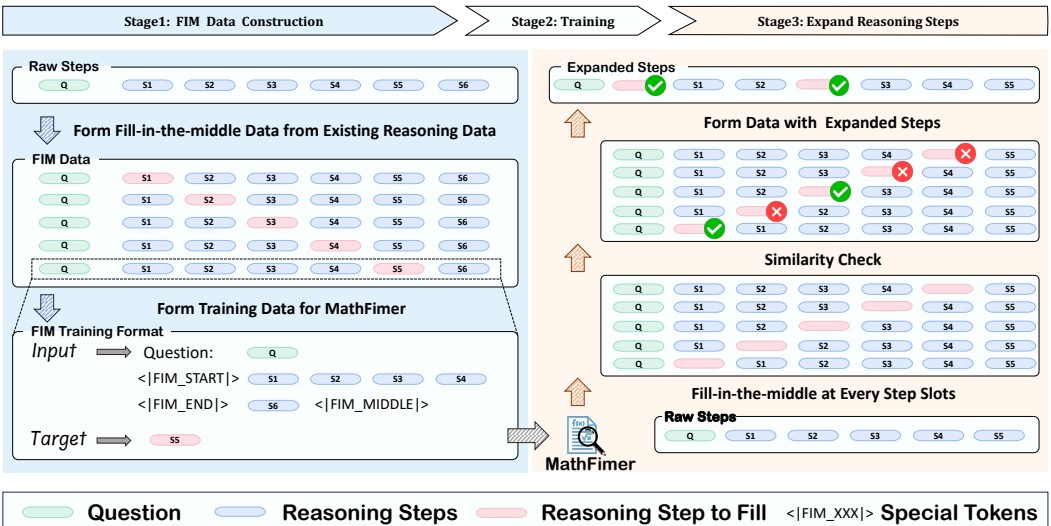

Figure 2: An overview of our work. The left part illustrates how we construct FIM training data from existing CoT data and train FIM models, MathFimer, which works on chain-of-thought. The right part demonstrates the process where MathFimer is used to expand the steps of existing CoT data for more detailed reasoning.

- We develop and release a specialized training dataset (NuminaMath-FIM) along with a step-expansion model MathFimer-7B, providing a practical and scalable solution for improving mathematical reasoning datasets.
- Through extensive experiments across multiple benchmarks and model architectures, we demonstrate that our approach consistently improves mathematical reasoning performance, offering new insights into the relationship between step granularity and reasoning quality in LLMs.

## 2 APPROACH

In this paper, we propose a reasoning step expansion method that enhances the quality of existing data by filling in possible missing steps at the step level. This is achieved through the fill-in-the-middle (FIM) task, which supplements existing CoT data. Specifically, the work presented in this paper can be divided into two parts: the first part involves training the aforementioned FIM models (Section 2.1), and the second part applies the trained FIM models to extend steps in existing data (Section 2.2). Figure 2 shows an overview of our work.

### 2.1 FIM MODEL TRAINING

The goal of this section is to train a fill-in-the-middle (FIM) model for mathematical reasoning tasks, which can generate the missing intermediate steps between a mathematical problem, its preceding steps, and its succeeding steps. This can be expressed as:

$$\textbf{FIM}(Q, P, S) \Rightarrow M, \tag{1}$$

where **FIM** refers to the model we are training, $Q$ (question) represents the mathematical problem, $P$ (prefix) refers to the preceding steps, $S$ (suffix) refers to the succeeding steps, and $M$ (middle) denotes the intermediate steps between $P$ and $S$.

We construct the data for training the FIM model using the existing high-quality mathematical reasoning dataset, NuminaMath-CoT. NuminaMath-CoT includes mathematical reasoning data of varying difficulty levels, containing 853K mathematical question-and-answer pairs, providing us with more generalizable data.

Specifically, we first performed a step-by-step decomposition of the NuminaMath-CoT data, transforming the standard answers into individual steps. We provide the details of step decomposition in

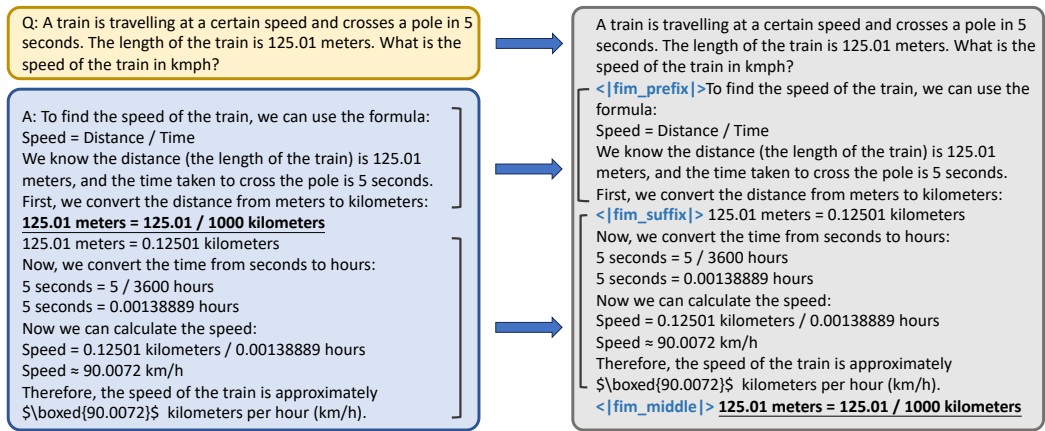

Figure 3: An example of NuminaMath-FIM. The left side represents a mathematical problem and its corresponding solution from NuminaMath-CoT, while the right side shows the FIM data constructed from it. The underlined portion represents a randomly selected step from all the steps, with the blue tokens <|fim_prefix|>, <|fim_suffix|>, and <|fim_middle|> being three special tokens. During supervised fine-tuning, we only compute the loss for the underlined portion.

Appendix D. Then, for each case, we randomly select one step and treat all the preceding steps as the prefix and all the succeeding steps as the suffix. This can be represented as:

$$(P, S, M) = (y_{1...i-1}, y_{i+1...n}, y_i), y_i \in Y \tag{2}$$

where $y_i$ is a step randomly selected from the answer $Y$, which contains $n$ steps. For the organization format of the FIM training data, we refer to the work of Bavarian et al. (2022) and adopt the PSM(Prefix-Suffix-Middle) sequence order. We use three special tokens: <|fim_prefix|>, <|fim_suffix|>, and <|fim_middle|>, to construct the format for the FIM training data. An example of the FIM data construction is provided in Figure 3.

For each case in NuminaMath-CoT, we performed three rounds of random sampling as described above. As a result, for each mathematical problem, we constructed three FIM data entries, which together formed our FIM training set, NuminaMath-FIM, consisting of 2.5M training samples for FIM task. Next, we conducted SFT on a math-specialized base model, Qwen2.5-Math-7B(Yang et al., 2024). Specifically, we only computed the loss for the tokens after <|fim_middle|>, ultimately obtaining the FIM model MathFimer-7B for step expansion.

## 2.2 EXPANSION OF REASONING STEPS

After training MathFimer-7B, we can use it to expand the reasoning steps in existing mathematical solutions. Specifically, for each pair of consecutive steps in the original solution, we perform an inference using the FIM model to generate potentially missing intermediate steps or provide more detailed reasoning between them. This can be formally expressed as follows:

$$\hat{y}_i = \mathbf{FIM}(Q, y_1 \ldots y_{i-1}, y_i \ldots y_n) \tag{3}$$

where i represents each position in the original answer, $n$ is the total number of steps in the original answer, $y_i$ is the i-th step in the original answer, **FIM** is the trained MathFimer model, $Q$ is the question for the sample, and $\hat{y}_i$ is the missing part generated by the FIM model between the i-th step and the subsequent steps.

In our experiments, we observed that when the original steps are already sufficiently detailed, the model tends to generate content that is very similar to the subsequent step $y_i$. Therefore, after the FIM model generates the supplementary step $\hat{y}_i$, we added a similarity calculation step. Specifically, we compute the sequence similarity between $\hat{y}_i$ and $y_i$. We set a threshold $\eta$ and mark those generated steps with a similarity greater than $\eta$ as invalid. In this paper, we set $\eta = 0.8$.

Next, we insert the steps generated by the FIM model into the original steps. Specifically, if the similarity score in the previous step is not labeled as invalid, we will insert it into the original

Table 1: Our main experimental results (%) on four mathematical reasoning tasks (GSM8K, MATH, Math Odyssey and OlympiadBench-EN). The evaluation results are obtained by sampling the model 16 times with a temperature of 0.7 and calculating the average accuracy.

| Dataset | FIM Model | Elementary Math | | Competition Math | | Average |
| --- | --- | --- | --- | --- | --- | --- |
| | | GSM8K | MATH | Odyssey | OB-EN | |
| **Base Model: Meta-Llama3.1-8B** | | | | | | |
| GSM8K+MATH | / | 67.55 | 18.32 | 21.59 | 1.78 | 27.31 |
| | MathFimer-7B | 73.16+5.61 | 21.84+3.52 | 21.34-0.25 | 2.52+0.74 | 29.72+2.41 |
| MathInstruct-CoT | / | 67.78 | 18.74 | 22.11 | 2.37 | 27.75 |
| | MathFimer-7B | 75.21+7.43 | 22.90+4.16 | 24.42+2.31 | 3.56+1.19 | 31.52+3.77 |
| MetaMathQA | / | 84.15 | 34.66 | 29.05 | 6.37 | 38.56 |
| | MathFimer-7B | 84.69+0.54 | 35.12+0.46 | 28.79-0.26 | 6.81+0.44 | 38.85+0.29 |
| **Base Model: Meta-Llama3.1-70B** | | | | | | |
| GSM8K+MATH | / | 89.23 | 40.22 | 38.30 | 8.74 | 44.12 |
| | MathFimer-7B | 92.72+3.49 | 44.36+4.14 | 37.79-0.51 | 12.15+3.41 | 46.76+2.63 |
| MathInstruct-CoT | / | 89.31 | 41.96 | 36.50 | 9.19 | 44.24 |
| | MathFimer-7B | 90.98+1.67 | 44.72+2.76 | 39.33+2.83 | 12.15+2.96 | 46.80+2.56 |
| MetaMathQA | / | 90.52 | 49.06 | 40.36 | 13.48 | 48.36 |
| | MathFimer-7B | 92.57+2.05 | 51.34+2.28 | 38.30-2.06 | 14.81+1.33 | 49.26+0.9 |
| **Base Model: Qwen2.5-Math-7B** | | | | | | |
| GSM8K+MATH | / | 82.71 | 50.90 | 36.25 | 15.41 | 46.32 |
| | MathFimer-7B | 85.37+2.66 | 51.92+1.02 | 34.7-1.55 | 14.37-1.04 | 46.59+0.27 |
| MathInstruct-CoT | / | 86.28 | 59.80 | 44.22 | 20.59 | 52.72 |
| | MathFimer-7B | 90.30+4.02 | 58.86-0.94 | 43.44-0.78 | 20.00-0.59 | 53.15+0.43 |
| MetaMathQA | / | 93.18 | 70.22 | 49.10 | 34.81 | 61.83 |
| | MathFimer-7B | 93.10-0.08 | 79.08+8.86 | 52.70+3.6 | 41.04+6.23 | 66.48+4.65 |
| **Base Model: Qwen2.5-Math-72B** | | | | | | |
| GSM8K+MATH | / | 93.25 | 70.74 | 50.13 | 30.37 | 61.12 |
| | MathFimer-7B | 94.24+0.99 | 75.16+4.42 | 52.70+2.57 | 36.30+5.93 | 64.60+3.48 |
| MathInstruct-CoT | / | 91.36 | 69.26 | 46.27 | 26.67 | 58.39 |
| | MathFimer-7B | 92.49+1.13 | 71.70+2.44 | 46.02-0.25 | 29.63+2.96 | 59.96+1.57 |
| MetaMathQA | / | 90.22 | 57.68 | 42.93 | 20.00 | 52.71 |
| | MathFimer-7B | 92.95+2.73 | 63.40+5.72 | 47.30+4.37 | 24.89+4.89 | 57.14+4.43 |

sequence. This insertion operation is carried out between each pair of original steps, ultimately constructing a more detailed answer with additional steps.

To evaluate the effectiveness and generalization of MathFimer-7B in expanding reasoning steps, we used it to extend the reasoning steps on several existing step-by-step reasoning datasets, including a mixture of GSM8K (Cobbe et al., 2021) and MATH (Hendrycks et al., 2021), MathInstruct-CoT (Yue et al., 2023), MetaMathQA (Yu et al., 2023), NuminaMath-CoT (Li et al., 2024), and ScaleQuestMath (Ding et al., 2025). For all datasets, we only used the training set. We conducted SFT on multiple base LLMs. For general-purpose LLMs, we selected Meta-Llama-3.1-8B/70B (Grattafiori et al., 2024), and for math-specialized LLMs, we chose Qwen2.5-Math-7B/72B (Yang et al., 2024). After SFT, we evaluated performance on multiple mathematical reasoning benchmarks, including GSM8K, MATH, Math Odyssey(Fang et al., 2025), and OlympiadBench-EN (He et al., 2024).

## 3 EXPERIMENTS

### 3.1 SETTINGS

We conducted supervised instruction fine-tuning experiments on both general-purpose and math-specialized foundation LLMs. We selected the original data before applying MathFimer-7B as the baseline for each experimental group and compared the performance improvements achieved after applying our proposed method for step expansion. In all experiments, we maintained identical training settings, only varying the data used for training. Specifically, we used Megatron-LM as the framework for SFT, with a model `max_length` set to 8k and a global batch size of 128 (GSM8K+MATH datasets were set to 32 due to their smaller sample sizes). The learning rate for training was set to 1e-5. We packed all training samples for faster training. All SFT experiments were conducted on 64 Ascend H910B-64G.

For evaluation, we employ vLLM(Kwon et al., 2023) as the inference framework. To reduce evaluation variance, each question is sampled 16 times with a temperature setting of 0.7, and the average accuracy is calculated. To determine whether the model-generated answers are correct, we utilize LLM-as-a-judge, thereby mitigating evaluation errors caused by answer extraction and rule-based comparison. All model inferences in this study are conducted on NVIDIA A100-80G GPUs, with 1-card inference for 7B/8B models and 4-cards inference for 70B/72B models.

### 3.2 MAIN RESULTS

We conducted our experiments on base models of different sizes, including both general-purpose and math-specialized models. Specifically, we evaluated Meta-Llama-3.1-8B, Meta-Llama-3.1-70B, Qwen2.5-Math-7B, and Qwen2.5-Math-72B. We employed the MathFimer-7B model, which was trained based on Qwen2.5-Math-7B, to perform a single round of step expansion. For comparative analysis, we selected five datasets: GSM8K+MATH, MathInstruct-CoT, MetaMathQA, NuminaMath-CoT, and ScaleQuest-Math, to examine whether step expansion via MathFimer-7B leads to improved performance on relevant mathematical reasoning benchmarks. For evaluation, we used the GSM8K, MATH, Math Odyssey, and OlympiadBench-EN datasets. Among them, GSM8K and MATH primarily assess elementary-level mathematical problems, while Math Odyssey and OlympiadBench-EN consist of competition-level mathematics questions.

We present all our main results in Table 1, and our full experimental results in Appendix K. As shown in the results, our method achieves consistent improvements across different base models and datasets. Specifically, for Meta-Llama3.1-8B, MathInstruct-CoT, when expanded using MathFimer, increases the average accuracy from 27.75% to 32.52%, yielding a 3.77 percentage point improvement. Similarly, for Qwen2.5-Math-72B, MetaMathQA, after step expansion via MathFimer, raises the average accuracy from 52.71% to 57.14%, achieving a gain of 4.43%.

From the experimental results, we observe that on certain models and specific datasets, applying MathFimer can lead to slight performance regressions. For example, on Qwen2.5-Math-7B, the MetaMathQA dataset exhibits a 0.08% performance drop after applying MathFimer. This occurs because introducing more detailed steps through MathFimer may occasionally introduce content that is difficult to fully control. We discuss these potential risks in the Limitations (in Appendix B) section. Nevertheless, when considering the overall average accuracy, MathFimer consistently improves performance across different base models. This demonstrates the practical effectiveness of MathFimer in enhancing the reasoning capabilities of LLMs.

Due to computational resource constraints, we perform **only a single round of step expansion** in our main experiment to observe the general applicability of our proposed MathFimer. However, MathFimer is capable of iterative step expansion, meaning that previously expanded steps can be further refined. We explore the scalability of step expansion in more detail in Section 4.2.

We provide examples of step expansion performed by MathFimer-7B on the GSM8K, MATH, and MathInstruct training sets in Appendix L. Each example contains two rounds of expansion: the first round expands upon the original CoT steps, while the second round further refines and elaborates on the reasoning steps based on the first-round output, resulting in more detailed reasoning trajectories.

# 4 ANALYSIS

## 4.1 DISENTANGLING MODEL EFFECTS

To disentangle the impact of our FIM methodology from model distillation effects, we conducted a systematic ablation study addressing a critical question: *To what extent do our performance gains stem from the FIM-based step expansion versus knowledge transfer from the base model?*

We designed a controlled experiment using Qwen2.5-Math-7B as the base model. We first generated distillation datasets by fine-tuning the base model on NuminaMath-CoT and using it to generate solutions for GSM8k+MATH, MathInstruct, and MetaMathQA. We then applied MathFimer-7B's step expansion to these distilled datasets to isolate the contribution of our FIM approach.

The results in Table 2 reveal several key insights. First, while distillation alone yields substantial improvements (e.g., MATH accuracy increases from 18.32% to 29.32% for G+M), MathFimer's step expansion provides additional gains even on distilled data (+3.28%). This pattern is consistent across datasets, with MI-CoT showing similar additive benefits (+2.88% on GSM8K). The smaller magnitude of improvements on distilled data compared to original data (e.g., +3.52 % vs +3.28 % for G+M on MATH) suggests that while knowledge transfer from the base model contributes significantly to overall performance, our FIM-based step expansion provides complementary benefits through structural enhancement of reasoning chains.

Table 2: Performance decomposition experimental results. Experiments are conduct on Meta-Llama-3.1-8B.

| Dataset | FIM Model | GSM8K | MATH | Odyssey | OB-EN |
|---|---|---|---|---|---|
| G+M | / | 67.55 | 18.32 | 21.59 | 1.78 |
| | 7B | 73.16$_{+5.61}$ | 21.84$_{+3.52}$ | 21.34$_{-0.25}$ | 2.52$_{+0.74}$ |
| G+M (distill) | / | 81.58 | 29.32 | 27.76 | 4.44 |
| | 7B | 82.41$_{+0.83}$ | 32.6$_{+3.28}$ | 28.19$_{+0.43}$ | 6.59$_{+2.15}$ |
| MI-CoT | / | 67.78 | 18.74 | 22.11 | 2.37 |
| | 7B | 75.21$_{+7.43}$ | 22.9$_{+4.16}$ | 24.42$_{+2.31}$ | 3.56$_{+1.19}$ |
| MI-CoT (distill) | / | 83.32 | 35.90 | 32.90 | 6.22 |
| | 7B | 86.2$_{+2.88}$ | 37.88$_{+1.98}$ | 32.85$_{-0.05}$ | 8.63$_{+2.41}$ |
| MMQA | / | 84.15 | 34.66 | 29.05 | 6.37 |
| | 7B | 84.69$_{+0.54}$ | 35.12$_{+0.46}$ | 28.79$_{-0.26}$ | 6.81$_{+0.44}$ |
| MMQA (distill) | / | 84.23 | 35.18 | 24.42 | 6.81 |
| | 7B | 87.57$_{+3.34}$ | 36.98$_{+1.8}$ | 26.16$_{+1.74}$ | 8.11$_{+1.3}$ |

## 4.2 ANALYSIS OF ITERATION EFFECTS

Our iterative step expansion experiments demonstrate the robust scalability of MathFimer. As shown in Table 3a, each iteration of step expansion consistently improves reasoning performance across most benchmarks. Notably, on the GSM8K benchmark, MI-CoT achieves substantial gains of +7.43%, +12.43%, and +15.54% percentage points over three iterations, reaching 83.32% accuracy. Similar patterns emerge on MATH, with consistent improvements culminating in a +9.42% percentage point gain.

This iterative enhancement suggests that MathFimer effectively constructs increasingly sophisticated reasoning chains, where each expansion cycle introduces valuable intermediate steps that contribute to improved reasoning capabilities. The consistent performance gains across different datasets and iteration counts validate the scalability of our approach and its ability to leverage extended reasoning chains for enhanced reasoning.

Table 3: Experimental Results of ablation studies.

| Dataset | Iter | GSM8K | MATH | Odyssey | OB-EN |
|---|---|---|---|---|---|
| G+M | 0 | 67.55 | 18.32 | 21.59 | 1.78 |
| | 1 | 73.16$_{+5.61}$ | 21.84$_{+3.52}$ | 21.34$_{-0.25}$ | 2.52$_{+0.74}$ |
| | 2 | 77.03$_{+9.48}$ | 23.50$_{+5.18}$ | 21.08$_{-0.51}$ | 6.07$_{+4.29}$ |
| | 3 | 78.7$_{+11.15}$ | 25.54$_{+7.22}$ | 22.37$_{+0.78}$ | 6.67$_{+4.89}$ |
| MI-CoT | 0 | 67.78 | 18.74 | 22.11 | 2.37 |
| | 1 | 75.21$_{+7.43}$ | 22.90$_{+4.16}$ | 24.42$_{+2.31}$ | 3.56$_{+1.19}$ |
| | 2 | 80.21$_{+12.43}$ | 26.68$_{+7.94}$ | 27.76$_{+5.65}$ | 4.44$_{+2.07}$ |
| | 3 | 83.32$_{+15.54}$ | 28.16$_{+9.42}$ | 26.48$_{+4.37}$ | 6.67$_{+4.3}$ |

(a) Ablation on iteration effects.

| Dataset | Size | GSM8K | MATH | Odyssey | OB-EN |
|---|---|---|---|---|---|
| G+M | / | 67.55 | 18.32 | 21.59 | 1.78 |
| | 1.5B | 73.09$_{+5.54}$ | 22.76$_{+4.44}$ | 21.59 | 1.78 |
| | 7B | 73.16$_{+5.61}$ | 21.84$_{+3.52}$ | 21.34$_{-0.25}$ | 2.52$_{+0.74}$ |
| | 72B | 73.09$_{+5.54}$ | 21.84$_{+3.52}$ | 23.39$_{+1.8}$ | 2.07$_{+0.29}$ |
| MI-CoT | / | 67.78 | 18.74 | 22.11 | 2.37 |
| | 1.5B | 73.01$_{+5.23}$ | 21.84$_{+3.1}$ | 22.62$_{+0.51}$ | 3.26$_{+0.89}$ |
| | 7B | 75.21$_{+7.43}$ | 22.90$_{+4.16}$ | 24.42$_{+2.31}$ | 3.56$_{+1.19}$ |
| | 72B | 73.92$_{+6.14}$ | 23.06$_{+4.32}$ | 24.68$_{+2.57}$ | 2.67$_{+0.3}$ |

(b) Ablation on different model size of MathFimer.

### 4.3 IMPACT OF MODEL SCALE

To investigate the relationship between model capacity and step expansion capability, we conducted a systematic comparison between MehtFimer-1.5B, MathFimer-7B and MathFimer-72B. We trained MathFimer-1.5B on Qwen2.5-Math-1.5B, MathFimer-72B on Qwen2.5-Math-72B using identical training data and hyperparameters as MathFimer-7B to ensure fair comparison.

Our experimental results, as presented in Table 3b, reveal an interesting finding: the performance gap between MathFimer-7B and MathFimer-72B is notably small across all benchmarks. For instance, on GSM8K+MATH, performance is nearly identical across all three model sizes (73.09%, 73.16%, and 73.09% on GSM8K). This pattern of comparable performance persists across different datasets and evaluation metrics, suggesting that step expansion quality may not be significantly bottlenecked by model capacity. These results indicate that the step expansion task might be effectively addressed with relatively modest model sizes, potentially due to the structured nature of mathematical reasoning steps and the explicit decomposition in our approach.

### 4.4 COMPARE WITH PROMPT-BASED FILL

We try to compare our method with prompt-based step expansion using general-purpose models, although we found it challenging to ensure a fair comparison. Prompt-based methods typically rely on external LLMs and repeated inference, which introduces additional computational costs and tuning complexity. In contrast, our approach leverages existing data and directly trains a FIM model, making step expansion more efficient and scalable without requiring external resources.

Table 4: Comparison with prompt-based fill method. Experiments are conduct on Meta-Llama-3.1-8B.

| Dataset | FIM Model | GSM8K | MATH | Odyssey | OB-EN |
|---------|-----------|-------|------|---------|-------|
| | / | 67.55 | 18.32 | 21.59 | 1.78 |
| G+M | MathFimer-1.5B | **73.09** | **22.76** | 21.59 | 1.78 |
| | Llama-3.2-3B-Instruct | 68.76 | 18.88 | **22.39** | **2.52** |
| | / | 67.78 | 18.74 | 22.11 | 2.37 |
| MI-CoT | MathFimer-1.5B | **73.01** | **21.84** | **22.62** | **3.26** |
| | Llama-3.2-3B-Instruct | 71.30 | 20.34 | 22.16 | 3.04 |

To provide a more concrete comparison, we conducted an ablation experiment leveraging prompt-based fill approach. Specifically, we prompted Llama-3.2-3B-Instruct in a zero-shot manner to expand intermediate steps in original answers, and perform SFT on this expanded dataset with the same settings of former experiments. As shown in Table 4, our approach consistently outperforms prompt-based expansion(on G+M, 73.09% vs. 68.76%), both in accuracy and quality of generated intermediate reasoning. This highlights the effectiveness of training specialized FIM models for automatic step enrichment. The detailed experimental settings are provided in Appendix E.

### 4.5 EVALUATION OF FILLED STEPS

To assess the correctness of the generated steps, we conducted a quantitative evaluation using Qwen2.5-Math-PRM-7B (Zhang et al., 2025b) as a process reward model. Specifically, we scored the reasoning steps both before and after expansion, and the results, reported as the proportion of steps achieving a PRM scores, are summarized in Figure 4.

Moreover, when comparing the PRM scores of expanded steps to those of the original reasoning chains, we observe that the correctness is largely preserved. In some cases, the expanded steps even outperform the original ones. These findings suggest that our step-expansion method does not degrade answer quality and,

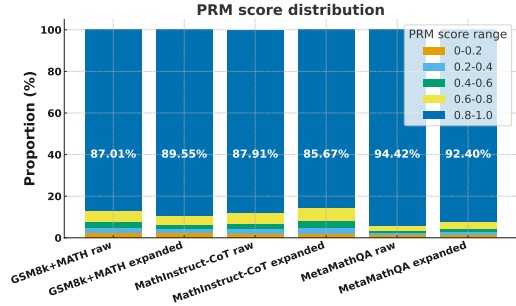

Figure 4: PRM score distribution before and after step insertion with MathFimer-7B. The PRM scores range from 0 to 1.

in fact, can improve the plausibility and completeness of intermediate reasoning. This supports the viability of our approach as a reliable step enhancement strategy for improving LLM reasoning.

Besides, we conducted a consistency check between human annotations and PRM scores. Specifically, we randomly sampled 100 examples from the MathFimer-expanded MathInstruct-CoT dataset

and had them manually evaluated by two authors of this paper. The agreement rate between human-annotated correct steps and those with a PRM score above 0.8 reached 94%. Interestingly, our findings suggest that the actual quality of the data generated by MathFimer may be even higher than what the PRM score indicates. In particular, we observed that some steps with relatively low PRM scores were still valid and helpful from a human evaluation perspective.

We additionally employ a PRM as a verification tool to filter the quality of steps produced by Math-Fimer. Experimental results show that filtering MathFimer-generated steps based on their PRM scores can further improve downstream task performance with carefully designed removal ratio. We provide a detailed description of this experiment and the corresponding analysis in Appendix J.

## 4.6 COMPARE WITH OTHER METHODS

To better showcase how MathFimer compares with other methods for enhancing reasoning in LLMs, we collected several representative approaches applied to the MetaMathQA benchmark, including Direct Preference Optimization(DPO), Step-DPO, Rejection Sampling Fine-tuning(RFT) and Proximal Policy Optimization(PPO). As shown in Table 5, our method achieves comparable performance under the same experimental settings. We summarize the comparative results of MathFimer and other methods as follows.

Table 5: Comparison with other methods for reasoning enhancement.

| Type | Method | Base Model | GSM8K | MATH |
|---|---|---|---|---|
| Fill-in-the-middle | MathFimer(ours) | Meta-Llama3.1-8B | 84.15 -> 86.58 (+2.43) | 34.66 -> 37.04 (+2.38) |
| Preference-based | DPO (Lai et al., 2024) | Qwen2-7B | unknown | 54.80 -> 55.00 (+0.20) |
| | Step-DPO (Lai et al., 2024) | Qwen2-7B | 88.20 -> 88.50 (+0.30) | 54.80 -> 55.80 (+1.00) |
| Rejection Sampling | RFT (Wang et al., 2024) | Mistral-7B | 77.90 -> 79.00 (+1.10) | 28.60 -> 29.30 (+1.30) |
| RL-based | PPO (Wang et al., 2024) | Mistral-7B | 77.90 -> 81.80 (+3.90) | 28.60 -> 31.30 (+2.70) |

- **Different optimization targets:** Methods like RL and DPO typically focus on optimizing for final answer correctness, while our approach specifically targets the quality and granularity of intermediate steps. These approaches are actually complementary rather than competitive, our expanded data could potentially serve as better starting points for RL.

- **Orthogonality and compatibility:** Our FIM-based approach can actually be used alongside RL and DPO. The expanded steps we generate could serve as higher-quality starting points for these optimization methods, potentially leading to even better results when combined.

- **Computational efficiency:** Our method is significantly more efficient than RL-based approaches, which require substantial computational resources for reward modeling and policy optimization. MathFimer can be applied using smaller models (even 1.5B parameters) with minimal overhead.

## 4.7 DOMAIN APPLICABILITY ANALYSIS

Motivated by the hypothesis that enriching reasoning steps improves an LLM's overall reasoning capability, we posit that the MathFimer framework can generalize beyond mathematics to other reasoning domains. To this end, we perform a domain applicability analysis consisting of two parts. In Section 4.7.1, we investigate whether a math-specific model trained on MathFimer-expanded data can also achieve improved performance on out-of-distribution non-mathematical reasoning tasks. In Section 4.7.2, we further examine whether FIM models trained on data from other domains can be used to expand reasoning steps in corresponding target domains, thereby evaluating the transferability of our proposed method.

## 4.7.1 EVALUATION ON OUT-OF-DISTRIBUTION TASKS

MathFimer expands CoT steps to induce models to produce more detailed reasoning traces. To verify whether such enhanced reasoning behaviors generalize to other domains, we evaluate models trained on MathFimer-expanded data in out-of-distribution (OOD) reasoning domains. Specifically, we use MathFimer-expanded MathInstruct-CoT as training data and fine-tune two base models, Qwen2.5-Math-1.5B and Qwen2.5-Math-7B. We evaluate these models on three OOD domains:

*general reasoning* using the BBH (Suzgun et al., 2023) dataset, which contains 27 subsets covering diverse complex reasoning tasks; *scientific reasoning* using GPQA_diamond (Rein et al., 2024) and MMLU_redux (Gema et al., 2025); and *logical reasoning* using LogicBenchBQA (Parmar et al., 2024) and HellaSwag (Zellers et al., 2019). The detailed results are reported in Table 6.

Table 6: Evaluation results (%) on OOD benchmarks.

| Base Model | FIM Model | General | Scientific | | Logic | |
| --- | --- | --- | --- | --- | --- | --- |
| | | BBH | GPQA_D | MMLU_R | LogicBench | hellaswag |
| Qwen2.5-Math-1.5B | / | 36.54 | 30.56 | 36.73 | 51.09 | 24.32 |
| | MathFimer-1.5B | 52.05 | 33.46 | 48.14 | 61.99 | 38.95 |
| Qwen2.5-Math-7B | / | 49.01 | 36.74 | 55.36 | 55.95 | 32.65 |
| | MathFimer-1.5B | 53.16 | 41.04 | 60.91 | 63.77 | 44.00 |

From our experimental results, we observe that MathFimer achieves substantial accuracy improvements across all reasoning benchmarks. In particular, when trained on Qwen2.5-Math-1.5B, it delivers approximately a 16% performance gain on the BBH dataset, demonstrating that the enhanced reasoning ability induced by MathFimer generalizes well to tasks in other reasoning domains.

### 4.7.2 Application of FIM Method on Other Domains

In addition to the OOD evaluations mentioned above, we conducted an additional experiment to verify that Fimer models can also be effectively trained using data from other domains to perform step expansion and improve downstream task performance. Specifically, we used natural reasoning data to construct FIM data and trained a ReasoningFimer-1.5B model based on Qwen2.5-1.5B. We then applied this model to expand steps in a logical-reasoning dataset, LogiCoT, and trained the resulting model. Finally, we evaluated it on the benchmarks described in Section 4.7.1, and the results are shown in Table 7.

Table 7: Evaluation results (%) of ReasoningFimer trained with general reasoning data.

| Base Model | FIM Model | General | Scientific | | Logic | |
| --- | --- | --- | --- | --- | --- | --- |
| | | BBH | GPQA_D | MMLU_R | LogicBench | hellaswag |
| Qwen2.5-1.5B | / | 37.92 | 29.42 | 51.74 | 64.49 | 39.97 |
| | ReasoningFimer-1.5B | 39.83 | 30.81 | 53.93 | 69.69 | 41.00 |
| Qwen2.5-7B | / | 37.09 | 33.21 | 47.93 | 46.63 | 40.98 |
| | ReasoningFimer-1.5B | 48.41 | 42.17 | 67.95 | 69.31 | 55.40 |

From our experimental results, we observe that the proposed step-expansion method is applicable to a broader range of domains. It consistently improves performance across reasoning benchmarks spanning multiple domains, thereby demonstrating the wide applicability of our approach. In addition, we argue that conventional reasoning tasks can leverage the MathFimer framework to perform step expansion and thereby generate more detailed reasoning traces. However, code reasoning tasks are less compatible with MathFimer. The primary reason is that existing datasets typically contain complete and executable code; inserting additional steps into such code would compromise its executability and thus degrade the quality of the resulting reasoning steps.

## 5 Conclusion

In this paper, we introduce the Fill-in-the-middle (FIM) paradigm into mathematical reasoning chains. We construct NuminaMath-FIM by decomposing solutions into prefix-suffix pairs, where intermediate steps are held out for reconstruction. Through training on these prefix-middle-suffix triplets, we develop MathFimer models that can effectively expand reasoning steps while preserving solution coherence. Our comprehensive experiments across multiple mathematical reasoning datasets demonstrate that MathFimer-enhanced data consistently improves model performance with relative improvements of 7.43% on GSM8K and 8.86% on MATH.

ACKNOWLEDGEMENT

This work was supported by National Natural Science Foundation of China (No. 62436007), National Natural Science Foundation of China (No. 62506332) and CIPS-LMG Huawei Innovation Fund.

ETHICS STATEMENT

This work does not involve human subjects, personal data, or sensitive information. All datasets used in our experiments are publicly available datasets designed for evaluating mathematical reasoning in LLMs. We strictly adhered to ethical research practices and did not conduct any data collection that could raise privacy, security, or fairness concerns. Our methods do not introduce risks of harmful applications. To the best of our knowledge, this research complies with the ICLR Code of Ethics and poses no foreseeable ethical concerns.

REPRODUCIBILITY STATEMENT

To facilitate reproducibility of our work, we provide a detailed step decomposition method (stated in Appendix D), along with the full set of training and evaluation hyperparameters (stated in Section 3.1). Since our training and evaluation procedures are standard and general, researchers can choose the training framework compatible with their hardware to train the model, and select the inference framework based on their device setup.

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

## A    LLM Usage Declaration

In writing this paper, we only used LLMs for polishing. The generation of ideas in this work **did not** involve any assistance from LLMs. The experimental design and manuscript writing were **not directly produced by LLMs** either. The models were used solely as a polishing tool: specifically, we first drafted the manuscript, then refined it with the help of an LLM, and finally the authors conducted another round of verification after polishing.

## B    Limitations

While our MathFimer framework demonstrates promising results in enhancing mathematical reasoning through step expansion, we identify several important limitations that warrant careful consideration and future investigation.

**Domain Generalization**    While our approach demonstrates effectiveness in mathematical reasoning, its applicability to other reasoning domains remains uncertain. The current implementation and evaluation focus exclusively on mathematical problem-solving, leaving open questions about the framework's generalizability to domains such as code reasoning, logical deduction, and common-sense reasoning, where solution structures and validation requirements may differ significantly.

**Generation Reliability**    Our step expansion process inherently relies on model generation, introducing potential risks of error propagation. Despite overall improvements in reasoning quality, we currently lack robust mechanisms for verifying the logical consistency and mathematical correctness of inserted steps. This limitation becomes particularly critical when applying multiple iterations of step expansion, where errors could potentially accumulate.

**Methodological Limitations**    The framework's effectiveness inherently depends on the quality of initial training data and may inherit biases from base models. Additionally, the current approach primarily focuses on expanding existing solution patterns rather than generating novel solution approaches, potentially limiting its applicability to extremely complex or unconventional problems.

**Data Applicability**    The core mechanism of MathFimer is to insert additional steps between insufficiently detailed CoT steps, thereby making the overall reasoning trajectory more comprehensive and coherent, which in turn enhances the model's reasoning capability. However, MathFimer may not be suitable for certain categories of data. First, for datasets whose CoT traces are already highly refined, such as R1-like reasoning trajectories, MathFimer may offer limited benefits, as the existing reasoning steps are sufficiently detailed. Moreover, MathFimer may also struggle with non-linear reasoning processes. For reasoning structures that are tree-shaped, graph-like, or contain refinement traces with significant leaps in reasoning, MathFimer may introduce steps that lack coherence, making the overall reasoning trajectory appear less natural.

## C    Related Works

### C.1    Mathematical Reasoning of LLMs

Mathematical reasoning is one of the advanced capabilities of large language models (LLMs). By transforming real-world mathematical problems into a sequence of sub-problems and engaging in step-by-step thinking, the model's ability to solve related mathematical tasks is enhanced (Wei et al., 2022). Currently, the mathematical reasoning ability of models can be strengthened at various stages of LLM's training. During the pre-training phase, reasoning-related knowledge texts, such as mathematical forum discussions, textbooks, and so on, are typically used for enhancement (Paster et al., 2023; Zhang et al., 2025a). Additionally, a large number of synthetic step-by-step reasoning question-answer pairs are used to train the model, allowing it to learn various reasoning patterns. In the instruction fine-tuning (SFT) phase, high-quality question-answer pairs are usually employed to help the model master the pattern of step-by-step thinking, thereby enabling it to solve reasoning problems (Ding et al., 2025; Zhou et al., 2024b). After SFT, researchers also use techniques such as

outcome supervision and process supervision to reinforce the model's mathematical reasoning process, ensuring that the model generates more accurate reasoning steps during inference (Lightman et al., 2023; Wang et al., 2024; Zhang et al., 2025b).

## C.2 EXPANSION OF REASONING STEPS

Just as the scaling law in model training applies, there is also a scaling law for LLMs during test-time. The former improves the model's reasoning ability by providing more training data (Hoffmann et al., 2022), while the latter increases the model's computational load during inference to enhance calculation accuracy, thereby improving performance (Brown et al., 2024; Snell et al., 2024). Expanding reasoning steps is one way to enhance the test-time computation of LLMs. By generating more detailed reasoning steps during inference, the model's reasoning performance can be improved.

There are several ways to expand reasoning steps. For example, in a training-free approach, prompts like Chain-of-Thought (Wei et al., 2022) can guide the model to perform more detailed reasoning. Using self-consistency (Wang et al., 2022) to perform multiple reasoning paths and vote on the most consistent answers is another option. Additionally, methods like tree-search combined with a verifier can be used to select the optimal reasoning path (Chen et al., 2024; Wan et al., 2024; Guan et al., 2025). On the other hand, training-based approaches involve transforming training data into more detailed steps (Jin et al., 2024; Ying et al., 2024) or incorporating behaviors like planning (Wang et al., 2023) and self-correction (Yan et al., 2025), which can increase the model's computation during test-time, thus improving reasoning performance.

## D    DETAILS OF STEP DECOMPOSITION

Following previous research (Lightman et al., 2023; Wang et al., 2024), in this paper, we constructed a detailed set of rules for step segmentation. These rules primarily divide steps based on natural language sentences, while additionally handling common mathematical elements such as formulas, making the steps more reasonable.

Our decomposition approach for NuminaMath-CoT solutions into individual steps employed a combination of rule-based parsing and mathematical structure recognition:

- **Step Identification:** We primarily used explicit step markers as boundaries (e.g., "Step 1:", "First", "Next", "Finally"). When these weren't present, we identified natural breakpoints in the reasoning through sentence boundaries that introduce new mathematical operations.
- **Mathematical Structure Parsing:** We parsed solutions to identify self-contained mathematical units, such as individual equation formations, algebraic manipulations, numerical computations, and logical deductions.
- **Granularity Control:** We ensured each step contained a single conceptual operation or transformation, avoiding steps that combined multiple reasoning actions.

Since our method involves expansion between steps, a more fine-grained segmentation approach allows for more reasonable expansion.

## E    FULL RESULTS OF PROMPT-BASED FILL METHOD

### E.1    PROMPT FOR ZEROSHOT PROMPT-BASED FILL METHODS

We provide the prompt we use to do zeroshot prompt-based fill with general-purpose LLMs in 5. In practice, {question} and {answer} are replaced with the actual reasoning problem and its corresponding solution.

## F    DATA STATISTICS AFTER STEP EXPANSION

To fully demonstrate the practical scalability of our proposed MathFimer, we report the data statistics before and after applying step expansion with MathFimer-7B. The statistics include the number

---

**Prompt for zero-shot prompt-based Fill**

I will give you a question and its answer, please make the answer more detailed. You can only insert new steps into the existing steps, DO NOT modify the existing steps. Please directly output the new answer.

Question: {question}

Answer: {answer}

---

Figure 5: Prompt for zero-shot prompt-based step fill.

of samples, average token count, total token count, average reasoning steps, and average PRM score. We use the Qwen2.5-Math-7B (Yang et al., 2024) tokenizer for token counting and Qwen2.5-Math-PRM-7B (Zhang et al., 2025b) for computing PRM scores. The results are summarized in Table 8. As shown, MathFimer is able to increase the number of reasoning steps while maintaining a relatively high PRM score, highlighting its strong performance.

Table 8: Data statistics before and after step expansion with MathFimer-7B.

| Dataset | FIM Model(iter) | Samples | # Tokens | $\sum$ Tokens | # Steps | PRM score |
|---------|-----------------|---------|----------|---------------|---------|-----------|
| G+M | / | 15K | 254.31 | 3.81M | 5.13 | 0.8578 |
| | MathFimer-7B (1) | 15K | 350.88+37.97% | 5.26M | 9.56+86.35% | 0.8732 |
| | MathFimer-7B (3) | 15K | 845.76+232.57% | 12.7M | 33.75+557.89% | 0.8678 |
| MI-CoT | / | 188K | 272.53 | 51.3M | 9.7 | 0.8764 |
| | MathFimer-7B (1) | 188K | 435.32+59.73% | 82M | 17.58+81.24% | 0.8912 |
| | MathFimer-7B (3) | 188K | 1228.94+350.94% | 231M | 61.21+531.03% | 0.8938 |
| MMQA | / | 395K | 245.41 | 96.93M | 8.81 | 0.9234 |
| | MathFimer-7B (1) | 395K | 387.19+57.77% | 152.94M | 16.57+88.08% | 0.9174 |
| | MathFimer-7B (3) | 395K | 1040.03+323.79% | 410.81M | 58.67+565.95% | 0.9208 |

## G  EXTENDED EXPERIMENTAL RESULTS

Here we provide the complete prompt-based step-expansion results in Table 9, including those obtained using MathFimer-1.5B, MathFimer-7B, and size-matched general-purpose models. In this experiment, all prompt-based general models are selected such that their parameter counts are greater than or equal to those of the corresponding MathFimer models, ensuring a fair comparison and demonstrating the effectiveness of MathFimer.

The experimental results show that training on data expanded by MathFimer leads to substantially better performance than using prompt-based step expansion.

## H  QUALITY EFFECTS OF INITIAL DATASET FOR MATHFIMER

In this work, we use NuminaMath-CoT, a high-quality mathematical reasoning dataset, to construct the training data for MathFimer. To ablate the impact of data quality on FIM performance, we conducted an additional experiment using MetaMathQA (Yu et al., 2023), a dataset of relatively lower quality compared to NuminaMath-CoT, to train the FIM model and followed the same downstream training procedure. We present the results of our ablation study in Table 10.

Our results demonstrate that higher-quality data (NuminaMath-CoT) is more beneficial for Math-Fimer, leading to stronger end-to-end performance improvements.

Table 9: Full results of experiments compared with prompt-based fill method.

| Dataset | FIM Model | GSM8K | MATH | Odyssey | OB-EN |
|---------|-----------|-------|------|---------|-------|
| GSM8K+MATH | / | 67.55 | 18.32 | 21.59 | 1.78 |
| | MathFimer-1.5B | **73.09** | **22.76** | 21.59 | 1.78 |
| | Llama-3.2-3B-Instruct | 68.76 | 18.88 | **22.39** | **2.52** |
| | Qwen2.5-3B-Instruct | 63.59 | 20.10 | 21.10 | 1.92 |
| GSM8K+MATH | / | 67.55 | 18.32 | **21.59** | 1.78 |
| | MathFimer-7B | **73.16** | 21.84 | 21.34 | **2.52** |
| | Llama-3.1-8B-Instruct | 69.52 | 21.08 | 21.34 | 1.78 |
| | Qwen2.5-7B-Instruct | 68.56 | **22.87** | 21.28 | 1.64 |
| MathInstruct-CoT | / | 67.78 | 18.74 | 22.11 | 2.37 |
| | MathFimer-1.5B | **73.01** | **21.84** | **22.62** | **3.26** |
| | Llama-3.2-3B-Instruct | 71.30 | 20.34 | 22.16 | 3.04 |

Table 10: Experimental results of training MathFimer with different initial datasets and using them for step expansion.

| Dataset | FIM Model | GSM8K | MATH |
|---------|-----------|-------|------|
| G+M | / | 67.55 | 18.32 |
| | MathFimer-1.5B (NuminaMath-CoT) | **73.09** | **22.76** |
| | MathFimer-1.5B (MetaMathQA) | 66.98 | 19.48 |
| MI-CoT | / | 67.78 | 18.74 |
| | MathFimer-1.5B (NuminaMath-CoT) | **73.01** | **21.84** |
| | MathFimer-1.5B (MetaMathQA) | 69.14 | 20.14 |

# I    REINFORCEMENT LEARNING WITH MATHFIMER

To verify the three points discussed in Section 5.6, we conducted an additional reinforcement learning (RL) experiment. Specifically, we performed RL on top of models that had been SFT-trained with either the original data or the MathFimer-expanded data.

For RL training, we used DAPO-Math-17K (Yu et al., 2025) as the dataset and adopted GRPO (DeepSeek-AI et al., 2025) as the optimization algorithm. The experiments were implemented with the veRL (Sheng et al., 2025) framework, using rule-based evaluation as the reward function. We set the initial learning rate to 1e-6, the training batch size to 512, the number of rollouts to 16, and the maximum response length to 4096. We carried out a total of 200 steps of RL training and evaluated the models on the GSM8K and MATH benchmarks. The results are presented in Table 11.

Table 11: Experimental results of RL training on the top of MathFimer.

| Dataset | RL Step | FIM Model | GSM8K | MATH |
|---------|---------|-----------|-------|------|
| GSM8K+MATH | 0 | / | 67.55 | 18.32 |
| | | MathFimer-1.5B | 73.09 | 22.76 |
| | 200 | / | 76.58$_{+9.03}$ | 29.66$_{+11.34}$ |
| | | MathFimer-1.5B | 84.14$_{+11.05}$ | 34.78$_{+12.02}$ |

From our experimental results, we can see that RL continues to yield consistent improvements when applied to our method, demonstrating the orthogonality between MathFimer and other reasoning-enhancement approaches. Moreover, models expanded with MathFimer exhibit greater relative performance gains after RL compared to before, indicating that MathFimer provides a stronger starting point for subsequent training stages such as RL.

## J    MATHFIMER WITH PRM VERIFIER

Because MathFimer's step expansion may occasionally introduce low-quality reasoning steps, such as redundant content or factual errors, we incorporate a verification mechanism into the expansion pipeline to isolate and ablate these potential sources of degradation. Specifically, we employ Qwen2.5-Math-PRM-7B (Zhang et al., 2025b) as a step-level verifier to score each step generated by MathFimer. During the merging phase, we filter steps according to their PRM scores by discarding the lowest-scoring 10%, 20%, and 30% of steps and retaining only the higher-quality ones for insertion. The resulting verified and expanded reasoning traces are then used for training. The corresponding experimental results are presented in Table 12.

Table 12: Experimental results of PRM score filtering.

| Dataset | FIM Model | GSM8K | MATH | Odyssey | OB-EN |
|---------|-----------|-------|------|---------|-------|
| GSM8K+MATH | / | 67.55 | 18.32 | 21.59 | 1.78 |
|  | MathFimer-1.5B (no filtered) | 73.09 | 22.76 | 21.59 | 1.78 |
|  | MathFimer-1.5B (10% lowest PRM score filtered) | **74.14** | **24.12** | **22.02** | **1.86** |
|  | MathFimer-1.5B (20% lowest PRM score filtered) | 73.56 | 22.88 | 21.77 | **1.86** |
|  | MathFimer-1.5B (30% lowest PRM score filtered) | 72.38 | 21.66 | 21.03 | 1.54 |

Our experimental results show that introducing a verification mechanism into the MathFimer step-expansion pipeline can improve downstream task performance to some extent. In this experiment, removing a subset of steps with relatively low PRM scores further enhances model performance. However, removing too many steps undermines the effectiveness of step expansion. For example, after filtering out 30% of the steps, the performance gain begins to drop. This trend is consistent with our statistics reported in the main paper: approximately 90% of the generated steps have high PRM scores in the range of 0.8–1.0. Once a portion of these high-quality steps is removed, the benefits of step expansion diminish.

## K    FULL EXPERIMENTS RESULT

We provide our full experimental results on NuminaMath-CoT and ScaleQuest-Math in Table 13. We additionally include in the table the training results of step expansion using MathFimer-7B on NuminaMath-CoT and ScaleQuest-Math.

Table 13: Our main experimental results (%) on four mathematical reasoning tasks (GSM8K, MATH, Math Odyssey and OlympiadBench-EN). The evaluation results are obtained by sampling the model 16 times with a temperature of 0.7 and calculating the average accuracy.

| Dataset | FIM Model | Elementary Math | | Competition Math | | AVERAGE |
|---------|-----------|-------|------|---------|-------|---------|
| | | GSM8K | MATH | Odyssey | OB-EN | |
| **Base Model: Meta-Llama3.1-8B** | | | | | | |
| GSM8K+MATH | / | 67.55 | 18.32 | 21.59 | 1.78 | 27.31 |
| | MathFimer-7B | $73.16_{+5.61}$ | $21.84_{+3.52}$ | $21.34_{-0.25}$ | $2.52_{+0.74}$ | $29.72_{+2.41}$ |
| MathInstruct-CoT | / | 67.78 | 18.74 | 22.11 | 2.37 | 27.75 |
| | MathFimer-7B | $75.21_{+7.43}$ | $22.9_{+4.16}$ | $24.42_{+2.31}$ | $3.56_{+1.19}$ | $31.52_{+3.77}$ |
| MetaMathQA | / | 84.15 | 34.66 | 29.05 | 6.37 | 38.56 |
| | MathFimer-7B | $84.69_{+0.54}$ | $35.12_{+0.46}$ | $28.79_{-0.26}$ | $6.81_{+0.44}$ | $38.85_{+0.29}$ |
| NuminaMath-CoT | / | 89.08 | 48.10 | 36.76 | 13.04 | 46.75 |
| | MathFimer-7B | $91.21_{+2.13}$ | $50.5_{+2.4}$ | $38.3_{+1.54}$ | $14.52_{+1.48}$ | $48.63_{+1.89}$ |
| ScaleQuest-Math | / | 91.21 | 59.52 | 38.82 | 20.74 | 52.57 |
| | MathFimer-7B | $91.05_{-0.16}$ | $59.56_{+0.04}$ | $40.36_{+1.54}$ | $21.63_{+0.89}$ | $53.15_{+0.58}$ |
| **Base Model: Meta-Llama3.1-70B** | | | | | | |
| GSM8K+MATH | / | 89.23 | 40.22 | 38.30 | 8.74 | 44.12 |
| | MathFimer-7B | $92.72_{+3.49}$ | $44.36_{+4.14}$ | $37.79_{-0.51}$ | $12.15_{+3.41}$ | $46.76_{+2.63}$ |
| MathInstruct-CoT | / | 89.31 | 41.96 | 36.50 | 9.19 | 44.24 |
| | MathFimer-7B | $90.98_{+1.67}$ | $44.72_{+2.76}$ | $39.33_{+2.83}$ | $12.15_{+2.96}$ | $46.8_{+2.56}$ |
| MetaMathQA | / | 90.52 | 49.06 | 40.36 | 13.48 | 48.36 |
| | MathFimer-7B | $92.57_{+2.05}$ | $51.34_{+2.28}$ | $38.3_{-2.06}$ | $14.81_{+1.33}$ | $49.26_{+0.9}$ |
| NuminaMath-CoT | / | 96.44 | 66.36 | 47.30 | 31.70 | 60.45 |
| | MathFimer-7B | $96.36_{-0.08}$ | $67.82_{+1.46}$ | $46.79_{-0.51}$ | $33.33_{+1.63}$ | $61.08_{+0.63}$ |
| ScaleQuest-Math | / | 94.24 | 74.02 | 52.44 | 35.70 | 64.10 |
| | MathFimer-7B | $95_{+0.76}$ | $74.42_{+0.4}$ | $49.36_{-3.08}$ | $36.89_{+1.19}$ | $63.92_{-0.18}$ |
| **Base Model: Qwen2.5-Math-7B** | | | | | | |
| GSM8K+MATH | / | 82.71 | 50.90 | 36.25 | 15.41 | 46.32 |
| | MathFimer-7B | $85.37_{+2.66}$ | $51.92_{+1.02}$ | $34.7_{-1.55}$ | $14.37_{-1.04}$ | $46.59_{+0.27}$ |
| MathInstruct-CoT | / | 86.28 | 59.80 | 44.22 | 20.59 | 52.72 |
| | MathFimer-7B | $90.3_{+4.02}$ | $58.86_{-0.94}$ | $43.44_{-0.78}$ | $20_{-0.59}$ | $53.15_{+0.43}$ |
| MetaMathQA | / | 93.18 | 70.22 | 49.10 | 34.81 | 61.83 |
| | MathFimer-7B | $93.1_{-0.08}$ | $79.08_{+8.86}$ | $52.7_{+3.6}$ | $41.04_{+6.23}$ | $66.48_{+4.65}$ |
| NuminaMath-CoT | / | 85.37 | 55.16 | 43.19 | 17.33 | 50.26 |
| | MathFimer-7B | $87.72_{+2.35}$ | $53_{-2.16}$ | $42.16_{-1.03}$ | $16.74_{-0.59}$ | $49.91_{-0.36}$ |
| ScaleQuest-Math | / | 93.78 | 70.52 | 50.13 | 34.81 | 62.31 |
| | MathFimer-7B | $93.86_{+0.08}$ | $79.38_{+8.86}$ | $54.24_{+4.11}$ | $40.44_{+5.63}$ | $66.98_{+4.67}$ |
| **Base Model: Qwen2.5-Math-72B** | | | | | | |
| GSM8K+MATH | / | 93.25 | 70.74 | 50.13 | 30.37 | 61.12 |
| | MathFimer-7B | $94.24_{+0.99}$ | $75.16_{+4.42}$ | $52.7_{+2.57}$ | $36.3_{+5.93}$ | $64.6_{+3.48}$ |
| MathInstruct-CoT | / | 91.36 | 69.26 | 46.27 | 26.67 | 58.39 |
| | MathFimer-7B | $92.49_{+1.13}$ | $71.7_{+2.44}$ | $46.02_{-0.25}$ | $29.63_{+2.96}$ | $59.96_{+1.57}$ |
| MetaMathQA | / | 90.22 | 57.68 | 42.93 | 20.00 | 52.71 |
| | MathFimer-7B | $92.95_{+2.73}$ | $63.4_{+5.72}$ | $47.3_{+4.37}$ | $24.89_{+4.89}$ | $57.14_{+4.43}$ |
| NuminaMath-CoT | / | 96.29 | 77.54 | 55.27 | 43.26 | 68.09 |
| | MathFimer-7B | $96.13_{-0.16}$ | $77.4_{-0.14}$ | $55.01_{-0.26}$ | $44.15_{+0.89}$ | $68.17_{+0.08}$ |
| ScaleQuest-Math | / | 94.09 | 80.22 | 54.24 | 44.30 | 68.21 |
| | MathFimer-7B | $94.47_{+0.38}$ | $80.82_{+0.6}$ | $55.27_{+1.03}$ | $43.7_{-0.6}$ | $68.57_{+0.35}$ |

## L    EXAMPLES OF STEP EXPANSION WITH MATHFIMER

In this section, we present illustrative examples demonstrating how MathFimer expands intermediate steps within the CoT process. These examples help elucidate the underlying mechanisms of MathFimer and highlight its effectiveness in enhancing LLM reasoning.

---

**GSM8K**

**Question:**

```
Betty is saving money for a new wallet which costs $100. Betty
 has only half of the money she needs. Her parents decided to
give her $15 for that purpose, and her grandparents twice as
much as her parents. How much more money does Betty need to
buy the wallet?
```

**Original CoT steps:**

```
In the beginning, Betty has only 100 / 2 = $<<100/2=50>>50.
Betty's grandparents gave her 15 * 2 = $<<15*2=30>>30.
This means, Betty needs 100 - 50 - 30 - 15 = $
<<100-50-30-15=5>>5 more.
```

**Expanded CoT steps (Iter 1):**

```
Betty's wallet costs $100.
In the beginning, Betty has only 100 / 2 = $<<100/2=50>>50.
Her parents gave her $15.
Betty's grandparents gave her 15 * 2 = $<<15*2=30>>30.
So, Betty has 50 + 30 + 15 = $<<50+30+15=95>>95.
This means, Betty needs 100 - 50 - 30 - 15 = $
<<100-50-30-15=5>>5 more.
```

**Expanded CoT steps (Iter 2):**

```
Let's calculate the total amount of money Betty has after
receiving money from her parents and grandparents.
Betty's wallet costs $100.
Her parents gave her $15.
In the beginning, Betty has only 100 / 2 = $<<100/2=50>>50.
Her grandparents gave her twice as much as her parents, so:
Her parents gave her $15.
Her grandparents gave her twice as much as her parents, so
they gave her 15 * 2 = $<<15*2=30>>30.
Betty's grandparents gave her 15 * 2 = $<<15*2=30>>30.
Now, let's add all the money she has:
So, Betty has 50 + 30 + 15 = $<<50+30+15=95>>95.
Now, she needs 100 - 95 = $<<100-95=5>>5 more to buy the
wallet.
This means, Betty needs 100 - 50 - 30 - 15 = $
<<100-50-30-15=5>>5 more.
```

---

**MATH**

**Question:**

```
Let \[f(x) = \left\{
\begin{array}{cl} ax+3, &\text{ if }x>2, \\
x-5 &\text{ if } -2 \le x \le 2, \\
```

```
2x-b &\text{ if } x <-2.
\end{array}
\right.\]
Find $a+b$ if the piecewise function is continuous (which
means that its graph can be drawn without lifting your pencil
from the paper).
```

**Original CoT steps:**

```
For the piecewise function to be continuous, the cases must "
meet" at $2$ and $-2$.
For example, $ax+3$ and $x-5$ must be equal when $x=2$.
This implies $a(2)+3=2-5$, which we solve to get $2a=-6 \
Rightarrow a=-3$.
Similarly, $x-5$ and $2x-b$ must be equal when $x=-2$.
Substituting, we get $-2-5=2(-2)-b$, which implies $b=3$.
So $a+b=-3+3=\boxed{0}$.
```

**Expanded CoT steps (Iter 1):**

```
We are told that $f$ is continuous.
For the piecewise function to be continuous, the cases must "
meet" at \$2\$ and \$-2\$.
This means that the two expressions must be equal when $x=2$
and $x=-2$.
For example, \$ax+3\$ and \$x-5\$ must be equal when \$x=2\$.
Thus, we have:
This implies \$a(2)+3=2-5\$, which we solve to get \$2a=-6 \
Rightarrow a=-3\$.
So $a=-3$.
Similarly, $x-5$ and $2x-b$ must be equal when $x=-2$.
This gives $-2-5=2(-2)-b$.
Substituting, we get $-2-5=2(-2)-b$, which implies $b=3$.
Therefore, $a=-3$ and $b=3$.
So $a+b=-3+3=\boxed{0}$.
```

**Expanded CoT steps (Iter 2):**

```
We have the function
```

$$f(x) = \begin{cases} ax+3, & \text{if } x > 2, \\ x-5 & \text{if } -2 \leq x \leq 2, \\ 2x-b & \text{if } x < -2. \end{cases}$$

```
We are told that $f$ is continuous.
We look at the "junction points" where the cases meet.
For the piecewise function to be continuous, the cases must "
meet" at \$2\$ and \$-2\$.
We set the cases equal to each other at these points.
This means that the two expressions must be equal when $x=2$
and $x=-2$.
We set the expressions equal to each other at these points.
For example, \$ax+3\$ and \$x-5\$ must be equal when \$x=2\$.
Similarly, $x-5$ and $2x-b$ must be equal when $x=-2$.
Thus, we have:
```

$$ax+3 = x-5 \text{ when } x = 2.$$

```
This implies \$a(2)+3=2-5\$, which we solve to get \$2a=-6 \
Rightarrow a=-3\$.
```

```
Also, ax + 3 and x − 5 must be equal when x = 2.
So $a=-3$.
For x = −2, we have x − 5 and 2x − b must be equal.
Similarly, $x-5$ and $2x-b$ must be equal when $x=-2$.
Thus, we have:
This gives $-2-5=2(-2)-b$.
Solving for b, we have −7 = −4 − b.
Substituting, we get $-2-5=2(-2)-b$, which implies $b=3$.
So b = 3.
Therefore, $a=-3$ and $b=3$.
We conclude:
So $a+b=-3+3=\boxed{0}$.
```

### MathInstruct

**Question:**

```
A psychiatrist has 4 patients that need 25 sessions in total.
One of the patients needs 6 sessions. Another patient needs 5
more than that. How many sessions would the remaining patients
 need?
```

**Original CoT steps:**

```
The second patient needs 6+5 = 11 sessions
25-11-6 = 8 sessions
The answer is 8
```

**Expanded CoT steps (Iter 1):**

```
If one patient needs 6 sessions and another needs 5 more than
that, then:
The second patient needs 6+5 = 11 sessions
So subtracting the sessions of the first two patients from the
total sessions needed:
25-11-6 = 8 sessions
The remaining patients need 8 sessions in total.
The answer is 8
```

**Expanded CoT steps (Iter 2), ~~strikethrough~~ indicates similarity-based filtering:**

```
The psychiatrist has 4 patients that need 25 sessions in
total.
If one patient needs 6 sessions and another needs 5 more than
that, then:
The first patient needs 6 sessions
The second patient needs 6+5 = 11 sessions
Together, these two patients need 6+11 = 17 sessions
So subtracting the sessions of the first two patients from the
total sessions needed:
25-6-11 = 8 sessions
25-11-6 = 8 sessions
Therefore, the remaining two patients would need a total of 8
sessions.
The remaining patients need 8 sessions in total.
The answer is 8
```

