# OpenReview forum: "MathFimer: Enhancing Mathematical Reasoning by Expanding Reasoning Steps through Fill-in-the-Middle Task"
_ICLR.cc/2026/Conference — ICLR 2026 Poster_

### Official Review · Reviewer_UmnE · 2025-10-24

**Soundness:** 3
**Presentation:** 3
**Contribution:** 2
**Rating:** 4
**Confidence:** 4

**Summary:**

This paper introduces MathFimer, a novel framework designed to enhance the mathematical reasoning capabilities of Large Language Models (LLMs) by improving the granularity and completeness of training data. The core methodology ingeniously adapts the "Fill-in-the-Middle" (FIM) task from the domain of code generation. The authors first train a specialized model, MathFimer-7B, on the NuminaMath-FIM dataset, which is created by decomposing existing high-quality mathematical solutions into prefix, middle, and suffix segments. This model learns to infer and generate missing intermediate reasoning steps.

Subsequently, the trained MathFimer model is applied to expand various existing mathematical reasoning datasets by inserting more detailed steps into their solution chains. The central claim of the paper is that fine-tuning LLMs on these MathFimer-expanded datasets leads to consistent performance improvements across multiple mathematical benchmarks (GSM8K, MATH, etc.) and base models (Llama-3.1, Qwen2.5-Math). The paper supports its claims with a comprehensive set of experiments, including various ablation studies.

**Strengths:**

- Novel Methodological Contribution: The adaptation of the Fill-in-the-Middle (FIM) paradigm to densify mathematical reasoning chains is a novel and insightful contribution. It shifts the focus from regenerating entire solutions to surgically enhancing existing ones, which is a more targeted and potentially more efficient approach to data augmentation for reasoning tasks.

- Practicality and Scalability: The framework offers a practical and scalable alternative to common data enhancement techniques that rely on expensive API calls to proprietary models or computationally intensive search algorithms. By training a moderately-sized FIM model, the authors demonstrate a path to improve open-source models in a more accessible and cost-effective manner. This is a significant practical contribution to the community.

- Comprehensive Experimental Validation: The paper is underpinned by extensive experiments. The authors demonstrate consistent performance gains across a diverse set of base models (of varying sizes and specializations) and evaluation benchmarks, which supports the generalizability and effectiveness of their method.

**Weaknesses:**

> Unconvincing Comparison to Prompt-Based Baselines and Its Implications:

The paper's claim of superiority over prompt-based expansion methods (Section 5.4) is undermined by a flawed and uninformative experimental design. The comparison is made between the proposed MathFimer model (a specialized model fine-tuned on a strong math-specific base) and a general-purpose small model (Llama-3.2-3B-Instruct) in a zero-shot setting. This baseline is not representative of what is achievable with current state-of-the-art prompting techniques.

The crucial question this experiment fails to answer is whether the proposed, resource-intensive approach of creating a dataset and training a specialized FIM model is truly necessary. It is highly plausible that a modern, powerful model (e.g., GPT-4, DeepSeek-V3, Qwen3) could, through simple zero-shot prompting, generate step expansions of similar or superior quality.

The experiment does not provide meaningful evidence to rule out this simpler, more direct alternative. Consequently, the justification for the entire MathFimer training pipeline is weakened. A compelling demonstration of the method's value would require comparing it against a strong prompt-based baseline, proving that the specialized FIM training provides benefits beyond what is already achievable with existing powerful models. Without such a comparison, the practical utility of the proposed method remains an open question.


> Unaddressed Risk of Error Propagation and Lack of Verification:

The FIM model can inevitably generate steps that contain factual, logical, or calculational errors. The paper acknowledges this but only employs a similarity filter, which is insufficient for verifying logical correctness. A single erroneous insertion can invalidate an entire reasoning chain, and this risk is compounded during iterative expansion. The lack of a robust verification mechanism is a major concern for the reliability of the generated data and the models trained on it. The paper would be strengthened by a qualitative analysis of failure cases and a discussion of potential mitigation strategies (e.g., using a verifier model).

> Unacknowledged Dependence on High-Quality Seed Data and Implicit Distillation:

The paper argues for its independence from powerful external models, and the ablation study in Table 2 effectively demonstrates that MathFimer's structural enhancement provides benefits orthogonal to standard knowledge distillation. This is a strong point.

However, the paper should still discuss the provenance of the MathFimer model's own capabilities. The model is trained on NuminaMath-CoT, a high-quality dataset whose solutions are likely collected by state-of-the-art models such as GPT-4. This implies that MathFimer's ability to generate high-quality intermediate steps is itself a form of implicit distillation—it learns to master the fine-grained reasoning patterns present in its teacher's data.

This does not invalidate the paper's findings, but it reframes the contribution. The work would be positioned more precisely as a highly efficient framework for "compiling" the detailed reasoning style of a powerful teacher into a reusable FIM model, which can then be used to structurally enhance solutions from various sources. Acknowledging this dependency provides a more complete picture of the method's place within the broader ecosystem of model improvement and highlights the critical importance of the initial seed dataset's quality.


> Ambiguity in the Step Decomposition Process:

The crucial pre-processing step of decomposing continuous solutions into discrete steps is based on heuristic rules (Appendix B) that lack specificity and may not be robust. The granularity of these steps directly influences the FIM training task and the final model's behavior. Different segmentation choices could lead to vastly different outcomes. The paper would benefit from a more detailed explanation of these rules, with examples, and a discussion of the sensitivity of the method to this decomposition process.


> Limited Applicability to Non-Linear Reasoning Paradigms:

The MathFimer framework is fundamentally built on the assumption of a linear, sequential Chain-of-Thought process. However, a new generation of powerful reasoning models (e.g., DeepSeek-R1, Qwen3) often exhibit non-linear, abstract, and "jumpy" reasoning paths that defy simple sequential decomposition. For such models, the concept of "filling in the middle" between two adjacent steps becomes ill-defined. This is a limitation, and the paper should discuss the applicability of its method in the context of these emerging, more complex reasoning architectures.

> Insufficient Analysis of PRM Score Distribution:

While Figure 4 is used to argue for the quality of the inserted steps, it also reveals a subtle but important trend: the proportion of "perfect" solutions (PRM score 0.8-1.0) sometimes decreases after expansion. A more nuanced discussion of this change in the score distribution would provide a more complete and balanced interpretation of the results.

**Questions:**

See Weakness

---

> ### Author Response · Authors · 2025-11-28
> **Rebuttal [1/4]**
>
> Dear Reviewer UmnE,
>
> Thank you for the effort and time you dedicated to reviewing our paper, and for your recognition of the contribution of our proposed methods. We have greatly benefited from your valuable feedback. Below are our responses to the weaknesses you pointed out:
>
> ---
>
> > **[Weakness 1]: Unconvincing Comparison to Prompt-Based Baselines and Its Implications
> The paper's claim of superiority over prompt-based expansion methods (Section 5.4) is undermined by a flawed and uninformative experimental design. The comparison is made between the proposed MathFimer model (a specialized model fine-tuned on a strong math-specific base) and a general-purpose small model (Llama-3.2-3B-Instruct) in a zero-shot setting. This baseline is not representative of what is achievable with current state-of-the-art prompting techniques.
> The crucial question this experiment fails to answer is whether the proposed, resource-intensive approach of creating a dataset and training a specialized FIM model is truly necessary. It is highly plausible that a modern, powerful model (e.g., GPT-4, DeepSeek-V3, Qwen3) could, through simple zero-shot prompting, generate step expansions of similar or superior quality.
> The experiment does not provide meaningful evidence to rule out this simpler, more direct alternative. Consequently, the justification for the entire MathFimer training pipeline is weakened. A compelling demonstration of the method's value would require comparing it against a strong prompt-based baseline, proving that the specialized FIM training provides benefits beyond what is already achievable with existing powerful models. Without such a comparison, the practical utility of the proposed method remains an open question.**
>
> Thank you for your question. We address your concerns from the following perspectives:
>
> * **MathFimer is not resource-intensive.** We emphasize that training MathFimer is computationally lightweight. The method does not require synthesizing new data; instead, we simply transform existing CoT data using deterministic rules, a process that typically completes within seconds. Moreover, training the FIM model on top of a 1.5B LLM incurs only minimal cost.
> * **Using a powerful model is prohibitively expensive.** While we agree that employing a powerful LLM for step filling may yield stronger performance, the associated cost is extremely high. For instance, applying MathFimer to MetaMathQA involves approximately 854M input tokens and 56M output tokens. Based on current GPT-4.1 pricing, the total cost would be
> $854 \times 3 + 56 \times 25 = \$3962.$This cost becomes even more prohibitive when multiple rounds of step expansion are required, as the token usage, and thus the expense, scales rapidly.
> * **Our experimental comparison is fair.** The prompt-based baselines presented in Section 4.4 and Appendix G provide a fair comparison. Specifically, we match the parameter scale of MathFimer and evaluate against general-purpose LLMs of equal or larger model size. This setup places MathFimer at a computational disadvantage; yet, MathFimer still consistently outperforms the prompt-based approach, demonstrating the effectiveness of our method.

---

> ### Author Response · Authors · 2025-11-28
> **Rebuttal [2/4]**
>
> > **[Weakness 2]: Unaddressed Risk of Error Propagation and Lack of Verification
> The FIM model can inevitably generate steps that contain factual, logical, or calculational errors. The paper acknowledges this but only employs a similarity filter, which is insufficient for verifying logical correctness. A single erroneous insertion can invalidate an entire reasoning chain, and this risk is compounded during iterative expansion. The lack of a robust verification mechanism is a major concern for the reliability of the generated data and the models trained on it. The paper would be strengthened by a qualitative analysis of failure cases and a discussion of potential mitigation strategies (e.g., using a verifier model).**
>
> Thank you for raising this point. **In Section 4.5 of the paper, we evaluate the quality of the steps generated by MathFimer.** Specifically, we employ Qwen2.5-Math-PRM-7B to score the expanded steps and observe that, **when comparing the PRM scores of expanded steps with those of the original CoT reasoning chains, the correctness of the expanded steps is largely preserved. In some cases, the expanded steps even achieve higher PRM scores than the original ones.**
>
> During the rebuttal phase, we further investigated the relationship between PRM scores and the actual quality of the reasoning steps through human evaluation. Concretely, we randomly sampled 100 examples from the MathFimer-expanded MathInstruct-CoT dataset and had them independently annotated by two authors of this paper. **The agreement between human-verified correct steps and those with PRM scores above 0.8 reached 94%.** Interestingly, our results suggest that the true quality of MathFimer-generated steps may even exceed what PRM scores imply. We found that certain steps with relatively low PRM scores were still valid and useful according to human judgment.
>
> We further explored whether step-level evaluation can be integrated into the MathFimer pipeline. To this end, we employed Qwen2.5-Math-PRM-7B as a step-level verifier to filter expanded steps. By adjusting a threshold, we removed steps with low PRM scores to prevent them from being inserted into the original reasoning chain. The detailed experimental results are presented in the following table.
>
> | Dataset | Filtering  | GSM8K | MATH | Odyssey | OB-EN |
> | -------- | -------- | -------- |-------- |-------- |-------- |
> | GSM8K+MATH     | no filtering     | 73.09  | 22.76 | 21.59 | 1.78 |
> |      | 10% lowest PRM score filtered     | 74.14  | 24.12 | 22.02 | 1.86 |
> |      | 20% lowest PRM score filtered     | 73.56  | 22.88 | 21.77 | 1.86 |
> |      | 30% lowest PRM score filtered     | 72.38  | 21.66 | 21.03 | 1.54 |
>
> Our experimental results show that **introducing a verification mechanism into the MathFimer step-expansion pipeline can improve downstream task performance to some extent**. These experiments and analyses have been added to Appendix J.

---

> ### Author Response · Authors · 2025-11-28
> **Rebuttal [3/4]**
>
> > **[Weakness 3]: Unacknowledged Dependence on High-Quality Seed Data and Implicit Distillation
> The paper argues for its independence from powerful external models, and the ablation study in Table 2 effectively demonstrates that MathFimer's structural enhancement provides benefits orthogonal to standard knowledge distillation. This is a strong point.
> However, the paper should still discuss the provenance of the MathFimer model's own capabilities. The model is trained on NuminaMath-CoT, a high-quality dataset whose solutions are likely collected by state-of-the-art models such as GPT-4. This implies that MathFimer's ability to generate high-quality intermediate steps is itself a form of implicit distillation—it learns to master the fine-grained reasoning patterns present in its teacher's data.
> This does not invalidate the paper's findings, but it reframes the contribution. The work would be positioned more precisely as a highly efficient framework for "compiling" the detailed reasoning style of a powerful teacher into a reusable FIM model, which can then be used to structurally enhance solutions from various sources. Acknowledging this dependency provides a more complete picture of the method's place within the broader ecosystem of model improvement and highlights the critical importance of the initial seed dataset's quality.**
>
> Thank you for your question. We would like to address it from two perspectives.
>
> **1. Quality of Seed Data.**
>
> As we stated in Appendix H, the effectiveness of MathFimer is constrained by the quality of the initial CoT data—higher-quality data lead to better step-expansion performance. Here are our opinions to this point:
>
> * **Quality of existing datasets:** LLMs have been developing rapidly, especially by 2025, where their reasoning capabilities have reached a new level. This progress has resulted in the emergence of numerous high-quality datasets. We believe that the current data quality is already sufficient for training Fimer models, thereby supporting the applicability of our method.
> * **Impact of data quality:** As shown in Appendix H, even when MathFimer is trained on data of slightly lower quality, we still observe noticeable performance gains. This demonstrates the robustness of MathFimer. As more high-quality datasets become available in the future, we expect MathFimer’s performance to further improve.
>
> **2. About distillation.**
>
> We acknowledge in Section 4.1 that training MathFimer on high-quality datasets may introduce confounding effects, and we conducted experiments to analyze this issue. After excluding the performance gains attributable to distillation, MathFimer still delivers consistent improvements, indicating that MathFimer indeed provides substantial benefits to downstream performance.
>
> In addition, during the rebuttal phase, we further conducted OOD experiments on benchmarks outside the category of math reasoning (see the new Section 4.7.1). In these experiments, we used a relatively low-quality dataset to construct the FIM data and trained an extremely small FIM model (1.5B). The results show that MathFimer still yields clear and consistent gains on OOD evaluations. This demonstrates that by encouraging models to produce more detailed reasoning processes, MathFimer improves overall reasoning ability in a domain-general manner, providing evidence that its effect goes beyond simple distillation.
>
> | **Base Model**           | **FIM Model**     | **BBH** | **GPQA_D** | **MMLU_R** | **LogicBench** | **hellaswag** |
> |--------------------------|-------------------|---------|------------|------------|----------------|---------------|
> | Qwen2.5-Math-1.5B        | /                 | 36.54   | 30.56      | 36.73      | 51.09          | 24.32         |
> |                          | MathFimer-1.5B    | 52.05   | 33.46      | 48.14      | 61.99          | 38.95         |
> | Qwen2.5-Math-7B          | /                 | 49.01   | 36.74      | 55.36      | 55.95          | 32.65         |
> |                          | MathFimer-1.5B    | 53.16   | 41.04      | 60.91      | 63.77          | 44.00         |
>
> From our experimental results, we observe that **MathFimer achieves substantial accuracy improvements across all reasoning benchmarks**. In particular, when trained on Qwen2.5-Math-1.5B, it delivers approximately a 16\% performance gain on the BBH dataset, demonstrating that **the enhanced reasoning ability induced by MathFimer generalizes well to tasks in other reasoning domains.**

---

> > ### Author Response · Authors · 2025-11-28
> > **Rebuttal [4/4]**
> >
> > > **[Weakness 4]: Ambiguity in the Step Decomposition Process
> > The crucial pre-processing step of decomposing continuous solutions into discrete steps is based on heuristic rules (Appendix B) that lack specificity and may not be robust. The granularity of these steps directly influences the FIM training task and the final model's behavior. Different segmentation choices could lead to vastly different outcomes. The paper would benefit from a more detailed explanation of these rules, with examples, and a discussion of the sensitivity of the method to this decomposition process.**
> >
> > Thank you for your question. In Appendix B, we provide a detailed description of our proposed step-segmentation method to facilitate reproducibility. Since our paper focuses primarily on mathematical reasoning, the segmentation strategy we adopt is aligned with existing practices in math-reasoning research [1-2], and we believe it is well-suited for this domain. The step-segmentation method used in Section 4.7.2 is identical to that in Appendix B, demonstrating that this approach remains effective even when applied to other domains.
> >
> > **The main contribution of MathFimer is to offer an easy-to-use paradigm for expanding reasoning steps.** We believe that different researchers can design domain-specific segmentation strategies tailored to their own data characteristics and tasks; MathFimer itself is robust with respect to the choice of segmentation method.
> >
> > Moreover, in Appendix L, we provide concrete examples illustrating how MathFimer performs step segmentation and step expansion in practice, enabling a clearer understanding of the proposed framework.
> >
> > Ref:
> > [1] Wang, et al. Math-Shepherd: Verify and Reinforce LLMs Step-by-step without Human Annotations.
> > [2] Lai, et al. Step-DPO: Step-wise Preference Optimization for Long-chain Reasoning of LLMs.
> >
> > > **[Weakness 5]: Limited Applicability to Non-Linear Reasoning Paradigms
> > The MathFimer framework is fundamentally built on the assumption of a linear, sequential Chain-of-Thought process. However, a new generation of powerful reasoning models (e.g., DeepSeek-R1, Qwen3) often exhibit non-linear, abstract, and "jumpy" reasoning paths that defy simple sequential decomposition. For such models, the concept of "filling in the middle" between two adjacent steps becomes ill-defined. This is a limitation, and the paper should discuss the applicability of its method in the context of these emerging, more complex reasoning architectures.**
> >
> > Thank you for pointing this out. We have added a new paragraph in the *Limitations* section to discuss the limitations of MathFimer when applied to this type of data.
> >
> > > **[Weakness 6]: Insufficient Analysis of PRM Score Distribution
> > While Figure 4 is used to argue for the quality of the inserted steps, it also reveals a subtle but important trend: the proportion of "perfect" solutions (PRM score 0.8-1.0) sometimes decreases after expansion. A more nuanced discussion of this change in the score distribution would provide a more complete and balanced interpretation of the results.**
> >
> > Thank you for pointing out this issue. In Section 4.5, we provide additional analysis regarding the behavior of PRM scores. Moreover, based on PRM scores, we conducted an additional experiment in which we filtered the generated steps according to their PRM scores. The corresponding results have already been presented in our response to Weakness 2, and we therefore do not repeat them here.
> >
> > ---
> >
> > Thank you again for the thorough review. We think that our rebuttal can address your main concern, if you have any further questions, we would be happy to answer them.
> >
> > Sincerely,
> > MathFimer Authors

---

### Official Review · Reviewer_vyPr · 2025-10-30

**Soundness:** 3
**Presentation:** 2
**Contribution:** 3
**Rating:** 6
**Confidence:** 2

**Summary:**

This paper introduces MathFimer, a novel and cost-effective method to enhance the mathematical reasoning capabilities of Large Language Models (LLMs). The core insight is that the quality and detail of intermediate reasoning steps (CoT) in training data fundamentally limit model performance. MathFimer addresses this by using a Fill-in-the-Middle (FIM) task to automatically expand and enrich existing CoT steps. By employing a smaller, specialized model for this data expansion, the method avoids the high computational cost associated with using large, powerful external models for data generation. The resulting expanded datasets are used to fine-tune state-of-the-art LLMs, leading to measurable performance improvements across standard mathematical reasoning benchmarks like GSM8K and MATH.

**Strengths:**

* The use of the Fill-in-the-Middle paradigm for explicitly expanding reasoning steps is a creative and new application.

* The experiments show consistent and significant performance improvements across various base models and diverse benchmarks.

**Weaknesses:**

* The expansion process is contingent on the correctness and structure of the initial CoT steps. If the original CoT contains logical errors or fundamentally flawed reasoning, the FIM process might only elaborate on the mistake, potentially creating overly confident but incorrect training data.

* While detailed steps are generally beneficial, the FIM approach risks generating reasoning chains that are unnecessarily verbose or contain redundant intermediate steps, which could increase inference latency and potentially confuse the target model.

* The paper would benefit from a more detailed qualitative analysis showing examples of how MathFimer fixes or expands weak steps versus how it handles already correct, concise steps.

**Questions:**

* How does MathFimer handle expansions that introduce factual errors (e.g., via human verification or auto-checks), and what error rate was observed in generated middles?


* Have you tested MathFimer on non-mathematical reasoning tasks (e.g., logical or commonsense reasoning)?

---

> ### Author Response · Authors · 2025-11-28
> **Rebuttal [1/3]**
>
> Dear Reviewer vyPr,
>
> Thank you for the effort and time you dedicated to reviewing our paper, and for your recognition of the contribution of our proposed methods. We have greatly benefited from your valuable feedback. Below are our responses to the weaknesses you pointed out:
>
> ---
>
> > **[Weakness 1]: The expansion process is contingent on the correctness and structure of the initial CoT steps. If the original CoT contains logical errors or fundamentally flawed reasoning, the FIM process might only elaborate on the mistake, potentially creating overly confident but incorrect training data.**
>
> Thank you for raising this question. As we stated in Appendix H, the effectiveness of MathFimer is constrained by the quality of the initial CoT data, higher-quality data lead to better step-expansion performance. We address your concern from two perspectives:
>
> * **Quality of existing datasets:** LLMs have been developing rapidly, especially by 2025, where their reasoning capabilities have reached a new level. This progress has resulted in the emergence of numerous high-quality datasets. We believe that the current data quality is already sufficient for training Fimer models, thereby supporting the applicability of our method.
> * **Impact of data quality:** As shown in Appendix H, even when MathFimer is trained on data of slightly lower quality, we still observe noticeable performance gains. This demonstrates the robustness of MathFimer. As more high-quality datasets become available in the future, we expect MathFimer’s performance to further improve.
>
> > **[Weakness 2]: While detailed steps are generally beneficial, the FIM approach risks generating reasoning chains that are unnecessarily verbose or contain redundant intermediate steps, which could increase inference latency and potentially confuse the target model.**
>
> Thank you for raising this question. We address this question from several perspectives:
>
> + **Redundant intermediate steps.** To mitigate redundancy introduced by step expansion, MathFimer incorporates a similarity-based filtering strategy within its pipeline. This mechanism effectively removes steps that are overly similar to existing ones, thereby reducing the likelihood of repeatedly generating near-duplicate reasoning steps.
> + **Inference latency.** Prior work such as DeepSeek-R1 has demonstrated that stronger reasoning capabilities often come with longer reasoning trajectories. These models rely on increased token-level computation—i.e., inference-time compute scaling—to solve more complex tasks. In scenarios involving challenging reasoning problems, we view inference latency as a secondary concern relative to achieving higher reasoning accuracy.
> + **Potential confusion introduced to the target model.** As discussed in Section 4.5, we incorporate PRM-based scoring to assess the quality of expanded steps. Empirically, the steps produced by MathFimer maintain, and in some cases even exceed, the quality of the original CoT traces. Therefore, the concern that step expansion might confuse the target model appears to be relatively minor in practice.
>
>
> > **[Weakness 3]: The paper would benefit from a more detailed qualitative analysis showing examples of how MathFimer fixes or expands weak steps versus how it handles already correct, concise steps.**
>
> Thank you for your suggestion. We have added examples in Appendix L illustrating how MathFimer expands CoT steps in practice, allowing readers to better understand the mechanism and effectiveness of MathFimer in real scenarios.

---

> > ### Author Response · Authors · 2025-11-28
> > **Rebuttal [2/3]**
> >
> > > **[Question 1]: How does MathFimer handle expansions that introduce factual errors (e.g., via human verification or auto-checks), and what error rate was observed in generated middles?**
> >
> > Thank you for raising this point. In Section 4.5 of the paper, we evaluate the quality of the steps generated by MathFimer. Specifically, we employ Qwen2.5-Math-PRM-7B to score the expanded steps and observe that, **when comparing the PRM scores of expanded steps with those of the original CoT reasoning chains, the correctness of the expanded steps is largely preserved. In some cases, the expanded steps even achieve higher PRM scores than the original ones.**
> >
> > During the rebuttal phase, we further investigated the relationship between PRM scores and the actual quality of the reasoning steps through human evaluation. Concretely, we randomly sampled 100 examples from the MathFimer-expanded MathInstruct-CoT dataset and had them independently annotated by two authors of this paper. **The agreement between human-verified correct steps and those with PRM scores above 0.8 reached 94%.** Interestingly, our results suggest that the true quality of MathFimer-generated steps may even exceed what PRM scores imply. We found that certain steps with relatively low PRM scores were still valid and useful according to human judgment.
> >
> > We further explored whether step-level evaluation can be integrated into the MathFimer pipeline. To this end, we employed Qwen2.5-Math-PRM-7B as a step-level verifier to filter expanded steps. By adjusting a threshold, we removed steps with low PRM scores to prevent them from being inserted into the original reasoning chain. The detailed experimental results are presented in the following table.
> >
> >
> > | Dataset | Filtering  | GSM8K | MATH | Odyssey | OB-EN |
> > | -------- | -------- | -------- |-------- |-------- |-------- |
> > | GSM8K+MATH     | no filtering     | 73.09  | 22.76 | 21.59 | 1.78 |
> > |      | 10% lowest PRM score filtered     | 74.14  | 24.12 | 22.02 | 1.86 |
> > |      | 20% lowest PRM score filtered     | 73.56  | 22.88 | 21.77 | 1.86 |
> > |      | 30% lowest PRM score filtered     | 72.38  | 21.66 | 21.03 | 1.54 |
> >
> > Our experimental results show that **introducing a verification mechanism into the MathFimer step-expansion pipeline can improve downstream task performance to some extent**. These experiments and analyses have been added to Appendix J.

---

> > > ### Author Response · Authors · 2025-11-28
> > > **Rebuttal [3/3]**
> > >
> > > > **[Question 2]: Have you tested MathFimer on non-mathematical reasoning tasks (e.g., logical or commonsense reasoning)?**
> > >
> > > Thank you for your question. We have added an analysis section in Section 4.7 to examine the domain applicability of MathFimer. Specifically, we conducted two analyses, which we introduce below.
> > >
> > > **1. Evaluation on Out-of-distribution Tasks**
> > >
> > > MathFimer expands CoT steps to induce models to produce more detailed reasoning traces. To verify whether such enhanced reasoning behaviors generalize to other domains, we evaluate models trained on MathFimer-expanded data in OOD reasoning domains. The detailed results are reported in the table below.
> > >
> > > | **Base Model**           | **FIM Model**     | **BBH** | **GPQA_D** | **MMLU_R** | **LogicBench** | **hellaswag** |
> > > |--------------------------|-------------------|---------|------------|------------|----------------|---------------|
> > > | Qwen2.5-Math-1.5B        | /                 | 36.54   | 30.56      | 36.73      | 51.09          | 24.32         |
> > > |                          | MathFimer-1.5B    | 52.05   | 33.46      | 48.14      | 61.99          | 38.95         |
> > > | Qwen2.5-Math-7B          | /                 | 49.01   | 36.74      | 55.36      | 55.95          | 32.65         |
> > > |                          | MathFimer-1.5B    | 53.16   | 41.04      | 60.91      | 63.77          | 44.00         |
> > >
> > > From our experimental results, we observe that **MathFimer achieves substantial accuracy improvements across all reasoning benchmarks**. In particular, when trained on Qwen2.5-Math-1.5B, it delivers approximately a 16\% performance gain on the BBH dataset, demonstrating that **the enhanced reasoning ability induced by MathFimer generalizes well to tasks in other reasoning domains.**
> > >
> > > **2. Application of FIM Method on Other Domains**
> > >
> > > In addition to the OOD evaluations mentioned above, we conducted an additional experiment to verify that Fimer models can also be effectively trained using data from other domains to perform step expansion and improve downstream task performance. The results are shown in the table below.
> > >
> > > | **Base Model**      | **FIM Model**          | **BBH** | **GPQA_D** | **MMLU_R** | **LogicBench** | **hellaswag** |
> > > |---------------------|-------------------------|--------:|-----------:|-----------:|---------------:|--------------:|
> > > | Qwen2.5-1.5B        | /                       | 37.92   | 29.42      | 51.74      | 64.49          | 39.97         |
> > > |                     | ReasoningFimer-1.5B     | 39.83   | 30.81      | 53.93      | 69.69          | 41.00         |
> > > | Qwen2.5-7B          | /                       | 37.09   | 33.21      | 47.93      | 46.63          | 40.98         |
> > > |                     | ReasoningFimer-1.5B     | 48.41   | 42.17      | 67.95      | 69.31          | 55.40         |
> > >
> > > We observe that **the proposed step-expansion method is applicable to a broader range of domains.** It consistently improves performance across reasoning benchmarks spanning multiple domains, thereby demonstrating the wide applicability of our approach. In addition, we argue that conventional reasoning tasks can leverage the MathFimer framework to perform step expansion and thereby generate more detailed reasoning traces. **However, code reasoning tasks are less compatible with MathFimer.** The primary reason is that existing datasets typically contain complete and executable code; inserting additional steps into such code would compromise its executability and thus degrade the quality of the resulting reasoning steps.
> > >
> > > ---
> > >
> > > Thank you again for the thorough review. We think that our rebuttal can address your main concern, if you have any further questions, we would be happy to answer them.
> > >
> > > Sincerely,
> > > MathFimer Authors

---

### Official Review · Reviewer_VryU · 2025-10-31

**Soundness:** 3
**Presentation:** 3
**Contribution:** 3
**Rating:** 8
**Confidence:** 4

**Summary:**

Mathematical reasoning in LLMs benefits from detailed, structured intermediate steps, but expanding these steps typically requires powerful teacher models or expensive search-based methods (e.g., MCTS). The authors introduce MathFimer, which leverages the FIM paradigm to automatically insert intermediate reasoning steps into existing solution chains without external supervision.

- They construct a large FIM training dataset by decomposing solutions from NuminaMath-CoT (853K examples) into prefix-suffix pairs, yielding around 2.5M FIM samples.
- They use this to train MathFimer-7B - a 7B-parameter model (based on Qwen2.5-Math-7B) to fill intermediate reasoning steps. This model is then used to expand reasoning datasets such as GSM8K, MATH, MetaMathQA, MathInstruct-CoT, and ScaleQuestMath.
- Given a question, prefix, and suffix of an existing solution, the model learns to generate the missing intermediate steps, effectively densifying the reasoning chain.
- Models fine-tuned on MathFimer-expanded datasets outperform those trained on original data across multiple benchmarks (GSM8K, MATH, Math Odyssey, OlympiadBench-EN)
- MathFimer achieves reasoning improvement without reliance on larger teacher models or computationally expensive inference (e.g., MCTS)

**Strengths:**

- The paper introduces a creative extension of the FIM objective, that was previously used in code completion, to mathematical reasoning. This adaptation is conceptually elegant and non-trivial because reasoning chains differ structurally from code.
- The idea of training a model for FIM - MathFimer - for inserting plausible intermediate steps into existing verified solutions is also elegant and computationally efficient.
- The experiments are extensive, covering multiple datasets (GSM8K, MATH, MetaMathQA, MathInstruct-CoT, ScaleQuestMath) and several base models (LLaMA-3.1-8B/70B, Qwen2.5-Math-7B/72B). The consistent improvements (typically +3–8%) across models lend credibility.
- The paper includes ablation studies disentangling FIM effects from distillation, exploring iteration effects, model scale, and prompt-based baselines. These analyses strengthen the attribution of gains to the proposed method.
- The paper reports detailed data construction, tokenization, thresholds, and training settings, enhancing reproducibility.
- Overall, the paper is clearly written and and offers a lucid read.

**Weaknesses:**

- While the paper demonstrates strong results in mathematical reasoning, the scope is narrowly confined to math. That FIM-based reasoning expansion could be a general mechanism for improving structured reasoning is not empirically validated beyond this domain.
- In table 1, there is drop in performance with MathFimer in some cases - this is not discussed.
- The evaluation relies heavily on LLM-as-a-judge for correctness and PRM (process reward models) for reasoning quality. While these are reasonable proxies, substantiating with some human evaluation will strengthen the work.
- The paper briefly acknowledges error accumulation in iterative expansion but doesn’t provide detailed evidence. Multi-round expansion might insert steps that subtly shift semantics or alter problem logic
- There is some repetition and redundancy in writing that could be avoided. For instance, the description of similarity calculation on page 5.
- "fill-in-the-middle" is introduced on page 2 without citation

**Questions:**

Covered above in the weaknesses

---

> ### Author Response · Authors · 2025-11-28
> **Rebuttal [1/2]**
>
> Dear Reviewer VryU,
>
> Thank you for the effort and time you dedicated to reviewing our paper, and for your recognition of the contribution of our proposed methods. We have greatly benefited from your valuable feedback. Below are our responses to the weaknesses you pointed out:
>
> ---
>
> > **[Weakness 1]: While the paper demonstrates strong results in mathematical reasoning, the scope is narrowly confined to math. That FIM-based reasoning expansion could be a general mechanism for improving structured reasoning is not empirically validated beyond this domain.**
>
> Thank you for your suggestion. We fully agree with your perspective: MathFimer should serve as a **general mechanism for improving reasoning ability**. Therefore, we added an analysis section in Section 4.7 to examine the domain applicability of MathFimer. Specifically, we conducted two analyses, which we introduce below.
>
> **1. Evaluation on Out-of-distribution Tasks**
>
> MathFimer expands CoT steps to induce models to produce more detailed reasoning traces. To verify whether such enhanced reasoning behaviors generalize to other domains, we evaluate models trained on MathFimer-expanded data in OOD reasoning domains. The detailed results are reported in the table below.
>
> | **Base Model**           | **FIM Model**     | **BBH** | **GPQA_D** | **MMLU_R** | **LogicBench** | **hellaswag** |
> |--------------------------|-------------------|---------|------------|------------|----------------|---------------|
> | Qwen2.5-Math-1.5B        | /                 | 36.54   | 30.56      | 36.73      | 51.09          | 24.32         |
> |                          | MathFimer-1.5B    | 52.05   | 33.46      | 48.14      | 61.99          | 38.95         |
> | Qwen2.5-Math-7B          | /                 | 49.01   | 36.74      | 55.36      | 55.95          | 32.65         |
> |                          | MathFimer-1.5B    | 53.16   | 41.04      | 60.91      | 63.77          | 44.00         |
>
> From our experimental results, we observe that **MathFimer achieves substantial accuracy improvements across all reasoning benchmarks**. In particular, when trained on Qwen2.5-Math-1.5B, it delivers approximately a 16\% performance gain on the BBH dataset, demonstrating that the enhanced reasoning ability induced by MathFimer generalizes well to tasks in other reasoning domains.
>
> **2. Application of FIM Method on Other Domains**
>
> In addition to the OOD evaluations mentioned above, we conducted an additional experiment to verify that Fimer models can also be effectively trained using data from other domains to perform step expansion and improve downstream task performance. The results are shown in the table below.
>
> | **Base Model**      | **FIM Model**          | **BBH** | **GPQA_D** | **MMLU_R** | **LogicBench** | **hellaswag** |
> |---------------------|-------------------------|--------:|-----------:|-----------:|---------------:|--------------:|
> | Qwen2.5-1.5B        | /                       | 37.92   | 29.42      | 51.74      | 64.49          | 39.97         |
> |                     | ReasoningFimer-1.5B     | 39.83   | 30.81      | 53.93      | 69.69          | 41.00         |
> | Qwen2.5-7B          | /                       | 37.09   | 33.21      | 47.93      | 46.63          | 40.98         |
> |                     | ReasoningFimer-1.5B     | 48.41   | 42.17      | 67.95      | 69.31          | 55.40         |
>
> We observe that **the proposed step-expansion method is applicable to a broader range of domains.** It consistently improves performance across reasoning benchmarks spanning multiple domains, thereby demonstrating the wide applicability of our approach. In addition, we argue that conventional reasoning tasks can leverage the MathFimer framework to perform step expansion and thereby generate more detailed reasoning traces. **However, code reasoning tasks are less compatible with MathFimer.** The primary reason is that existing datasets typically contain complete and executable code; inserting additional steps into such code would compromise its executability and thus degrade the quality of the resulting reasoning steps.

---

> > ### Author Response · Authors · 2025-11-28
> > **Rebuttal [2/2]**
> >
> > > **[Weakness 2]: In table 1, there is drop in performance with MathFimer in some cases - this is not discussed.**
> >
> > Thank you for pointing this out. In the revision, we have added an analysis of the performance drop in Section 3.2.
> >
> > > **[Weakness 3]: The evaluation relies heavily on LLM-as-a-judge for correctness and PRM (process reward models) for reasoning quality. While these are reasonable proxies, substantiating with some human evaluation will strengthen the work.**
> >
> > Thank you for your valuable suggestion. In the past few days, we conducted a consistency check between human annotations and PRM scores. Specifically, we randomly sampled 100 examples from the MathFimer-expanded MathInstruct-CoT dataset and had them manually evaluated by two authors of this paper. **The agreement rate between human-annotated correct steps and those with a PRM score above 0.8 reached 94%.** Interestingly, our findings suggest that the actual quality of the data generated by MathFimer may be even higher than what the PRM score indicates. In particular, we observed that some steps with relatively low PRM scores were still valid and helpful from a human evaluation perspective. We have added this information to Section 4.5 in our revision.
> >
> >
> > > **[Weakness 4]: The paper briefly acknowledges error accumulation in iterative expansion but doesn’t provide detailed evidence. Multi-round expansion might insert steps that subtly shift semantics or alter problem logic**
> >
> > Thank you for your question. Since MathFimer relies on an LLM to generate reasoning steps, and we cannot fully control the correctness of all intermediate steps produced by the LLM, error accumulation may occur. We analyze and discuss this issue from several perspectives:
> >
> > + In our revision, **we added Appendix L, which presents concrete examples of step expansion performed by MathFimer.** From these examples, we observe that in the second iteration on MathInstruct, the model generated duplicated reasoning steps, but our similarity-based filtering mechanism effectively removed them.
> > + In Section 4.5, **we used a PRM to score the steps generated by MathFimer.** The presence of steps with low PRM scores suggests that MathFimer inevitably produces some erroneous steps.
> >
> > Given these considerations, we conducted an additional experiment during the rebuttal period. Specifically, we used Qwen2.5-Math-PRM-7B as a verifier to filter the expanded steps. We controlled a threshold and removed the steps with low PRM scores to prevent them from being inserted into the original reasoning chain. The detailed experimental results are shown in the following table.
> >
> >
> > | Dataset | Filtering  | GSM8K | MATH | Odyssey | OB-EN |
> > | -------- | -------- | -------- |-------- |-------- |-------- |
> > | GSM8K+MATH     | no filtering     | 73.09  | 22.76 | 21.59 | 1.78 |
> > |      | 10% lowest PRM score filtered     | 74.14  | 24.12 | 22.02 | 1.86 |
> > |      | 20% lowest PRM score filtered     | 73.56  | 22.88 | 21.77 | 1.86 |
> > |      | 30% lowest PRM score filtered     | 72.38  | 21.66 | 21.03 | 1.54 |
> >
> > Our experimental results show that **introducing a verification mechanism into the MathFimer step-expansion pipeline can improve downstream task performance to some extent**. These experiments and analyses have been added to Appendix J.
> >
> > > **[Weakness 5]: There is some repetition and redundancy in writing that could be avoided. For instance, the description of similarity calculation on page 5.**
> >
> > We reviewed the statements in the paper and revised several parts of the description. We also adjusted the redundancy on page 5. Thank you for pointing out this issue and helping us further polish our paper.
> >
> > > **[Weakness 6]: "fill-in-the-middle" is introduced on page 2 without citation**
> >
> > We have added this citation, and we appreciate you for pointing out this issue.
> >
> > ---
> >
> > Thank you again for the thorough review. We think that our rebuttal can address your main concern, if you have any further questions, we would be happy to answer them.
> >
> > Sincerely,
> > MathFimer Authors

---

### Official Review · Reviewer_UHHz · 2025-11-02

**Soundness:** 3
**Presentation:** 3
**Contribution:** 3
**Rating:** 6
**Confidence:** 3

**Summary:**

The authors propose MathFimer, a novel framework that enhances mathematical reasoning in large language models (LLMs) by expanding intermediate reasoning steps using a Fill-in-the-Middle (FIM) approach. Inspired by code completion tasks, MathFimer trains a model to insert missing or more detailed steps into existing chain-of-thought (CoT) solutions. The method is applied to various math datasets (e.g., GSM8K, MATH), and the resulting expanded data is used to fine-tune both general-purpose and math-specialized LLMs, leading to consistent performance improvements.

**Strengths:**

1. The use of FIM for step expansion in mathematical reasoning is innovative and offers a scalable alternative to expensive methods like MCTS or distillation from larger models.
2. Extensive experiments across multiple datasets and model sizes show consistent improvements (e.g., +7.43% on GSM8K, +8.86% on MATH), demonstrating the method’s effectiveness.
3. The approach works well even with smaller models (e.g., 1.5B), and supports iterative expansion, making it computationally efficient and widely applicable.

**Weaknesses:**

1. The method is only evaluated on mathematical reasoning. Its applicability to other domains (e.g., logic, code, commonsense reasoning) remains unclear.
2. The expanded steps are generated by a model and not verified for correctness or logical consistency, which may introduce errors, especially after multiple iterations.
3. The effectiveness of MathFimer heavily relies on the quality of the initial CoT data. Poor-quality base solutions could limit or mislead the expansion process.

**Questions:**

1. Could you incorporate a verification mechanism (e.g., a verifier or symbolic solver) to validate the correctness of the expanded reasoning steps?
2. Have you tested MathFimer on non-mathematical reasoning tasks? What are the challenges or adaptations needed to apply FIM-based expansion to other structured reasoning domains like code or legal reasoning?
3. What is the risk of over-expansion? Could inserting too many intermediate steps lead to redundancy or confusion? Have you analyzed the trade-off between step granularity and reasoning clarity?

---

> ### Author Response · Authors · 2025-11-28
> **Rebuttal [1/2]**
>
> Dear Reviewer UHHz,
>
> Thank you for the effort and time you dedicated to reviewing our paper, and for your recognition of the contribution of our proposed methods. We have greatly benefited from your valuable feedback. Below are our responses to the weaknesses you pointed out:
>
> ---
>
> > **[Weakness 1]: The method is only evaluated on mathematical reasoning. Its applicability to other domains (e.g., logic, code, commonsense reasoning) remains unclear.
> > [Question 2]: Have you tested MathFimer on non-mathematical reasoning tasks? What are the challenges or adaptations needed to apply FIM-based expansion to other structured reasoning domains like code or legal reasoning?**
>
> Thank you for your suggestion. We added an analysis section in Section 4.7 to examine the domain applicability of MathFimer. Specifically, we conducted two analyses, which we introduce below.
>
> **1. Evaluation on Out-of-distribution Tasks**
>
> MathFimer expands CoT steps to induce models to produce more detailed reasoning traces. To verify whether such enhanced reasoning behaviors generalize to other domains, we evaluate models trained on MathFimer-expanded data in OOD reasoning domains. The detailed results are reported in the table below.
>
> | **Base Model**           | **FIM Model**     | **BBH** | **GPQA_D** | **MMLU_R** | **LogicBench** | **hellaswag** |
> |--------------------------|-------------------|---------|------------|------------|----------------|---------------|
> | Qwen2.5-Math-1.5B        | /                 | 36.54   | 30.56      | 36.73      | 51.09          | 24.32         |
> |                          | MathFimer-1.5B    | 52.05   | 33.46      | 48.14      | 61.99          | 38.95         |
> | Qwen2.5-Math-7B          | /                 | 49.01   | 36.74      | 55.36      | 55.95          | 32.65         |
> |                          | MathFimer-1.5B    | 53.16   | 41.04      | 60.91      | 63.77          | 44.00         |
>
> From our experimental results, we observe that **MathFimer achieves substantial accuracy improvements across all reasoning benchmarks**. In particular, when trained on Qwen2.5-Math-1.5B, it delivers approximately a 16\% performance gain on the BBH dataset, demonstrating that **the enhanced reasoning ability induced by MathFimer generalizes well to tasks in other reasoning domains.**
>
> **2. Application of FIM Method on Other Domains**
>
> In addition to the OOD evaluations mentioned above, we conducted an additional experiment to verify that Fimer models can also be effectively trained using data from other domains to perform step expansion and improve downstream task performance. The results are shown in the table below.
>
> | **Base Model**      | **FIM Model**          | **BBH** | **GPQA_D** | **MMLU_R** | **LogicBench** | **hellaswag** |
> |---------------------|-------------------------|--------:|-----------:|-----------:|---------------:|--------------:|
> | Qwen2.5-1.5B        | /                       | 37.92   | 29.42      | 51.74      | 64.49          | 39.97         |
> |                     | ReasoningFimer-1.5B     | 39.83   | 30.81      | 53.93      | 69.69          | 41.00         |
> | Qwen2.5-7B          | /                       | 37.09   | 33.21      | 47.93      | 46.63          | 40.98         |
> |                     | ReasoningFimer-1.5B     | 48.41   | 42.17      | 67.95      | 69.31          | 55.40         |
>
> We observe that **the proposed step-expansion method is applicable to a broader range of domains.** It consistently improves performance across reasoning benchmarks spanning multiple domains, thereby demonstrating the wide applicability of our approach. In addition, we argue that conventional reasoning tasks can leverage the MathFimer framework to perform step expansion and thereby generate more detailed reasoning traces. **However, code reasoning tasks are less compatible with MathFimer.** The primary reason is that existing datasets typically contain complete and executable code; inserting additional steps into such code would compromise its executability and thus degrade the quality of the resulting reasoning steps.

---

> > ### Author Response · Authors · 2025-11-28
> > **Rebuttal [2/2]**
> >
> > > **[Weakness 2]: The expanded steps are generated by a model and not verified for correctness or logical consistency, which may introduce errors, especially after multiple iterations.
> > > [Question 1]: Could you incorporate a verification mechanism (e.g., a verifier or symbolic solver) to validate the correctness of the expanded reasoning steps?**
> >
> > Thank you for raising this point. We acknowledge that MathFimer may encounter issues of error introduction during step expansion. Given these considerations, we conducted an additional experiment during the rebuttal period. Specifically, **we used Qwen2.5-Math-PRM-7B as a verifier to filter the expanded steps.** We controlled a threshold and removed the steps with low PRM scores to prevent them from being inserted into the original reasoning chain. The detailed experimental results are shown in the following table.
> >
> >
> > | Dataset | Filtering  | GSM8K | MATH | Odyssey | OB-EN |
> > | -------- | -------- | -------- |-------- |-------- |-------- |
> > | GSM8K+MATH     | no filtering     | 73.09  | 22.76 | 21.59 | 1.78 |
> > |      | 10% lowest PRM score filtered     | 74.14  | 24.12 | 22.02 | 1.86 |
> > |      | 20% lowest PRM score filtered     | 73.56  | 22.88 | 21.77 | 1.86 |
> > |      | 30% lowest PRM score filtered     | 72.38  | 21.66 | 21.03 | 1.54 |
> >
> > Our experimental results show that **introducing a verification mechanism into the MathFimer step-expansion pipeline can improve downstream task performance to some extent**. These experiments and analyses have been added to Appendix J.
> >
> >
> > > **[Weakness 3]: The effectiveness of MathFimer heavily relies on the quality of the initial CoT data. Poor-quality base solutions could limit or mislead the expansion process.**
> >
> > Thank you for raising this question. As we stated in Appendix H, the effectiveness of MathFimer is constrained by the quality of the initial CoT data—higher-quality data lead to better step-expansion performance. We address your concern from two perspectives:
> >
> > * **Quality of existing datasets:** LLMs have been developing rapidly, especially by 2025, where their reasoning capabilities have reached a new level. This progress has resulted in the emergence of numerous high-quality datasets. We believe that the current data quality is already sufficient for training Fimer models, thereby supporting the applicability of our method.
> > * **Impact of data quality:** As shown in Appendix H, even when MathFimer is trained on data of slightly lower quality, we still observe noticeable performance gains. This demonstrates the robustness of MathFimer. As more high-quality datasets become available in the future, we expect MathFimer’s performance to further improve.
> >
> >
> >
> > > **[Question 3]: What is the risk of over-expansion? Could inserting too many intermediate steps lead to redundancy or confusion? Have you analyzed the trade-off between step granularity and reasoning clarity?**
> >
> > We appreciate your insightful question. To address redundancy within the pipeline, we incorporate a similarity-based filtering mechanism that ensures newly introduced reasoning steps do not replicate existing content. Regarding potential confusion, we employ a PRM-based evaluator to score generated steps. Interestingly, the model-produced steps achieve even higher scores than the original ones (Section 4.5). While MathFimer is proposed as a general paradigm for step expansion, we believe it can be further enhanced, for example, by integrating stronger verification mechanisms, as suggested by our preliminary experiment in Appendix J, which could yield additional performance gains.
> >
> > As for analyzing step granularity and reasoning clarity, such investigations require extensive human annotation and systematic evaluation. At present, we do not have the resources to perform such studies. We look forward to future work from the research community that explores these important dimensions of LLM reasoning.
> >
> > ---
> >
> > Thank you again for the thorough review. We think that our rebuttal can address your main concern, if you have any further questions, we would be happy to answer them.
> >
> > Sincerely,
> > MathFimer Authors

---

### Author Response · Authors · 2025-12-02
**Summary of Rebuttal & Discussion Phases**

Dear Reviewers, AC, SAC, and PC,

We sincerely appreciate the time and effort you devoted to reviewing our work, as well as the insightful feedback you provided, which has significantly strengthened our paper. Below, we summarize the progress so far.

---
We are encouraged to see that the reviewers consistently recognized several key strengths of our work, including:

* **Innovative and elegant method:** The use of FIM for step expansion in mathematical reasoning is innovative and elegant. (Reviewer `UHHz`, `VryU`, `vyPr`, `UmnE`)
* **Strong and consistent experimental results:** Through extensive experiments, MathFimer consistently and significantly improves mathematical reasoning of language models compared to the baseline. (Reviewer `UHHz`, `VryU`, `vyPr`, `UmnE`)
* **Efficient and widely applicable method:** MathFimer offers a practical and scalable method to enhance mathematical reasoning capabilities of LLMs with low costs. (Reviewer `UHHz`, `VryU`, `vyPr`, `UmnE`)
* **Comprehensive experiments and analyses:** The paper offers extensive implementation details, supplementary experiments, and analyses in both the main text and the appendix to thoroughly evaluate the proposed MathFimer method. (Reviewer `VryU`, `UmnE`)


---
During the rebuttal phase, we conducted additional experiments or supplemented relevant analyses in the revision to address the reviewers' concerns:

+ In Section 4.7, we introduce a comprehensive analysis of **MathFimer’s domain applicability**. Specifically, Section 4.7.1 evaluates the MathFimer-enhanced model on a suite of general, logical, and scientific reasoning benchmarks, while Section 4.7.2 trains a *ReasoningFimer* using general-purpose reasoning data to expand inference steps. Across both settings, **MathFimer yields consistent performance gains, demonstrating its strong cross-domain transferability.** (Reviewer `UHHz`, `VryU`, `vyPr`)
+ Section 4.5 now provides **a more detailed examination of the PRM score with human evaluation**, and Appendix J presents experiments that employ Qwen2.5-Math-PRM-7B as a verifier to filter MathFimer-generated steps. The results indicate that **applying quality filtering leads to further improvements, offering a principled pathway for more rigorous and reliable deployment of MathFimer.** (Reviewer `UHHz`, `VryU`, `vyPr`, `UmnE`)
+ Section 3.2 has been expanded with additional analyses of the main results, and the Limitations section now includes a discussion on data applicability, **enabling a more precise understanding of when and how our method is most effective.** (Reviewer `VryU`, `UmnE`)
+ Appendix L **provides three concrete examples of MathFimer performing step expansion**, offering readers a clear and intuitive view of the method’s operational behavior and its impact on reasoning trajectories. (Reviewer `VryU`, `vyPr`, `UmnE`)

---
We also clarified several key points to address the reviewers' concerns:

+ We provide a more thorough discussion of the data used to train MathFimer, showing that even relatively low-quality datasets can yield measurable gains. As data quality improves, the performance improvements introduced by MathFimer become increasingly pronounced.
+ We analyze potential issues that may arise during the step-expansion process, such as low-quality inserted steps or duplicated reasoning steps, and show that our existing mitigation strategies substantially reduce their adverse effects.
+ We further examine how MathFimer may influence model behavior, noting that more fine-grained reasoning traces can increase inference latency, a trade-off inherent to producing more detailed CoT trajectories.
+ We discuss the applicability of the Step Decomposition Process across different domains and argue that MathFimer offers a generalizable mechanism for enhancing LLM reasoning. Empirical results demonstrate its strong cross-domain transferability.

---
Once again, we sincerely thank all reviewers for their recognition and valuable suggestions. We believe that our responses address the raised concerns effectively, and we hope the program committee (reviewers, ACs, SACs, PCs) will take them into account in the next phases.

Sincerely,
MathFimer Authors

---

### Meta-Review · Area_Chair_euSL · 2026-01-02

**Summary:**

This paper proposes a method named MathFimer, which imitates the fill-in-the-middle task in code models to mathematical reasoning. The authors first train a specific model to do the fill-in-the-middle task, and then use this model to expand the existing CoT data to a more enhanced version. The expanded data is used for fine-tuning the final math model for better reasoning. Extensive experiments demonstrate the effectiveness of the method.

The reviewers raise several concerns from different views. As a summary, most of the concerns are addressed in my eyes. However, some of the concerns are not well addressed, but I think they will not affect the decision of the paper. The paper ideally introduces a methodology that can be extended and applied along the development of the LLM's reasoning ability.

Therefore, I recommend accept for this work.

**Reviewer Concerns:**

One common concern is about the generalization of the method, the author provided necessary additional experiments on OOD settings to demonstrate the robustness and universal of the method.
Also, for the concern about the correctness of the inserted step and the initial correctness of the existing CoTs (the initial quality of the CoTs), the authors conduct ablation studies to investigate the impact on different initial CoTs, and the experiments show the robustness of the work.
For other small questions, the authors give necessary rebuttal which I think the concerns are addressed.

However, as for the concerns from Reviewer UmnE, the authors may not give satisfied responses, such as the prompt-based baselines, and the distillation concern. The rebuttal somehow skips the direct answer and gives a discussion.

**Reviewer Scores:**

In my view, the Reviewer UHHz, VryU and vyPr may not change their score since the initial assessment is already positive and the concerns are easy to address. However, for Reviewer UmnE, it is possible to change the decision to a borderline accept, though the rebuttal is not so satisfied.

---

### Decision · Program_Chairs · 2026-01-26

Accept (Poster)